# Signals from the head and germinative region differentially regulate regeneration competence of the tapeworm *Hymenolepis diminuta*

Elise McCollough Nanista[1], Landon Elizabeth Poythress[1], Isabell Reese Skipper[1], Trevor Haskins[1], Marieher Felix Cora[2] and Tania Rozario[1,*]

## ABSTRACT

Competence to regenerate lost tissues varies widely across species. The rat tapeworm *Hymenolepis diminuta* undergoes continual cycles of shedding and regenerating thousands of reproductive segments to propagate the species. Despite its prowess, *H. diminuta* can only regenerate posteriorly from a singular tissue: the neck or germinative region (GR). What cells and signaling pathways restrict regeneration competence to the GR? In this study, we show that the head regulates regeneration competence by promoting maintenance of the GR and inhibiting proglottid formation in a distance-dependent manner. Anterior-posterior patterning within the GR provides local signals that contribute to these responses. *βcat1* is necessary for stem cell maintenance, proliferation and proglottidization, and *sfrp* is necessary for maintaining the GR at its proper length. Our study demonstrates that the head organizes a balance of pro- and anti-regeneration signals that must be integrated together and therefore control competence to regenerate.

KEY WORDS: Regeneration, Tapeworm, Planaria, Flatworm, Wnt signaling

## INTRODUCTION

Regeneration occurs over a wide range of biological scales from regeneration of axons at the cellular scale to whole body regeneration in planarians and *Hydra* (Bely and Nyberg, 2010). Tapeworms represent a fascinating case study in regeneration. They descend from a monophyletic clade (Neodermata) exclusively populated by parasites within the Platyhelminthes phylum (Laumer et al., 2015). As a sister group to planarians, it is possible that shared regenerative abilities have enabled tapeworms to achieve their considerable growth potential and reproductive prowess. Many species of intestinal tapeworms shed large pieces of their body to disperse the developing embryos for consumption by an intermediate host, while continuously replenishing the lost tissue. Studies with the rat tapeworm *Hymenolepis diminuta* have demonstrated that anterior fragments regenerate reproductive segments (proglottids) after serial rounds of amputation and transplantation into new host intestines for over a decade (Goodchild, 1958; Read, 1967). This greatly prolongs the normal lifespan of *H. diminuta*, suggesting that this parasite may not inherently need to age and die.

In a previous study, by culturing *H. diminuta in vitro* we were able to determine its range of regenerative abilities (Rozario et al., 2019) (Fig. 1A). Head regeneration was never observed. Posterior regeneration of new proglottids occurred but only when the neck/germinative region (GR) was retained. Neither the head nor proglottids could regenerate new proglottids. Hence, the GR is the only regeneration-competent tissue. We confirmed that, similar to the serial transplantation experiment described above (Read, 1967), 2 mm anterior fragments of *H. diminuta* regenerated proglottids after serial rounds of amputation and growth *in vitro*. However, proglottid regeneration was finite when the head region was removed, indicating that persistent regeneration does require head-dependent signals.

Proglottid regeneration depends on GR-resident stem cells, but this population is not functionally unique. Upon a lethal dose of irradiation that depletes stem cells in the GR, proglottid regeneration was inhibited (Rozario et al., 2019). However, the GR-resident stem cells were replaceable. Transplantation of donor cells into irradiated GRs rescued regeneration, as long as cycling stem cells were present in the donor pool (Rozario et al., 2019). Interestingly, the rescue occurred regardless of whether the donor cells came from GRs or from regeneration-incompetent parts of the body (Rozario et al., 2019). Hence, while stem cells are necessary for regeneration, signals within the GR are paramount.

A plethora of unknown signals likely operate in the GR and control stem cell proliferation, survival and differentiation, as well as proglottidization and competence to regenerate. Here, we show that regeneration competence is tightly linked to signals from the head. The head plays seemingly contradictory roles as it is necessary to maintain the GR but negatively regulates proliferation and proglottid regeneration. We find two genes typically associated with Wnt signaling, *βcat1* and *sfrp*, to be essential mediators of stem cell maintenance and GR length, respectively. Our study demonstrates that a balance of pro- and anti-regeneration signals must be overcome to enable GR regeneration and then initiation of proglottids. *H. diminuta* provides a fascinating example of how extrinsic signals can promote or restrict the ability to regenerate.

## RESULTS

### *H. diminuta* can regenerate the GR

Anterior fragments containing the head, GR and a few immature proglottids are competent to regenerate new proglottids

[1]University of Georgia, Athens, GA 30602, USA. [2]University of Puerto Rico-Cayey, PR 00736, USA.

*Author for correspondence (tania.rozario@uga.edu)

E.M.N., 0009-0002-3844-876X; L.E.P., 0009-0007-5836-5714; I.R.S., 0009-0009-1869-7292; T.H., 0009-0007-1528-3807; M.F.C., 0009-0004-9971-0826; T.R., 0000-0002-9971-5211

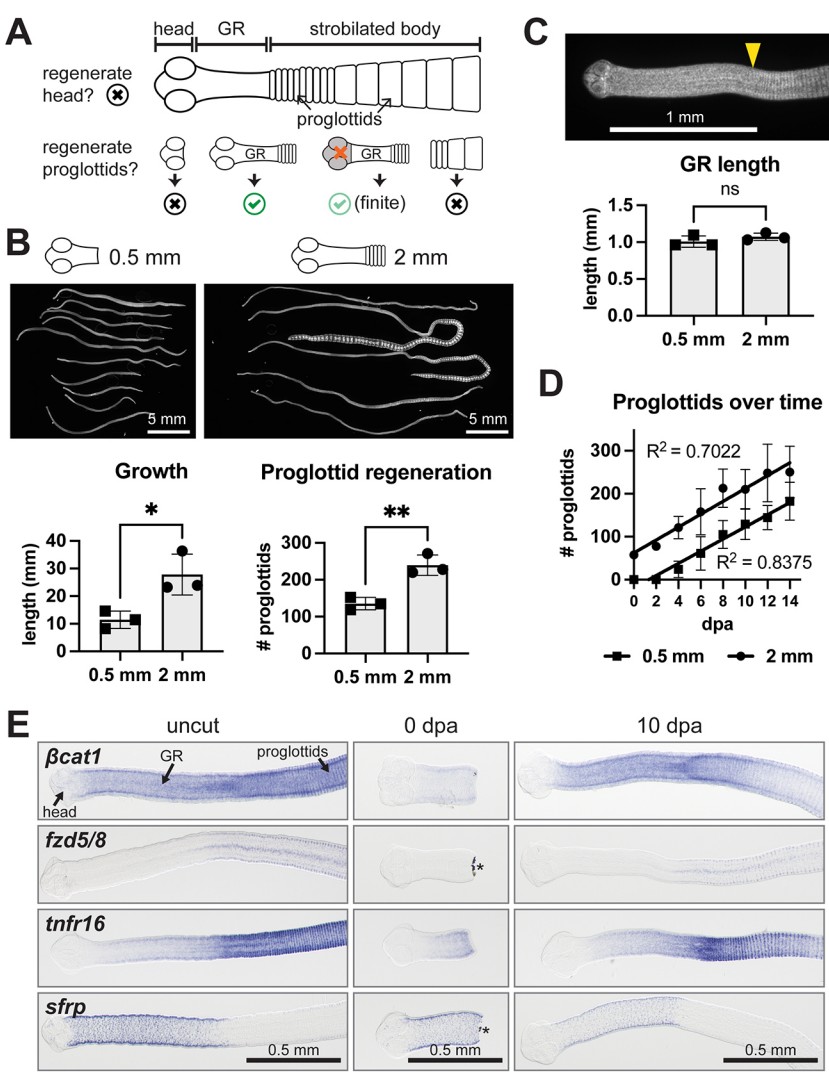

**Fig. 1. The germinative region regenerates after amputation.** (A) Schematic of basic *H. diminuta* anatomy and known regenerative ability. Head regeneration does not occur but proglottids regenerate from fragments that retain the germinative region (GR). Without the head region, proglottid regeneration initially occurs but subsequently ceases. (B) DAPI-stained anterior fragments cut within the GR (0.5 mm) or outside of it (2 mm) at 15 dpa. Quantification of worm length and proglottids regenerated at 15 dpa. $N=3$, $n=23$, 27; *$P<0.05$, **$P<0.01$ (unpaired, two-tailed *t*-test). (C) Top: DAPI-stained 0.5 mm anterior regenerate at 15 dpa. Yellow arrowhead marks the first proglottid. Bottom: Quantification of GR lengths from B; unpaired, two-tailed *t*-test. ns, not significant. (D) Quantification of proglottids regenerated every 2 days. Linear regression fitted; $n=50$, 55. (E) WISH for transcripts with A-P polarized expression patterns 3 days after acclimation *in vitro* (uncut) then amputated at 0.5 mm and stained at 0 dpa and 10 dpa. Anterior toward the left. Asterisks mark pigmented debris. Error bars represent s.d.

(Rozario et al., 2019). However, it is unclear whether the GR can regenerate itself and subsequently proglottidize. After acclimating worms to *in vitro* culture conditions for 3 days, we amputated within the GR (to obtain 0.5 mm fragments) or outside the GR (2 mm fragments) and grew the anterior fragments for 15 days. In both cases, proglottids regenerated (Fig. 1B). At 15 days post-amputation (dpa), GR lengths in regenerates from both groups were comparable at 1.01±0.08 mm and 1.07±0.05 mm (Fig. 1C). The 0.5 mm fragments regenerated GRs within 2-4 days before new proglottids were added (Fig. 1D). The rate of proglottid regeneration was comparable between both fragment types, at 14-15 proglottids per day (Fig. 1D). Thus, when amputated within the GR, *H. diminuta* is capable of regenerating the posterior GR and subsequently proglottids.

The GR is patterned by differentially expressed genes along the anterior-posterior (A-P) axis (Rozario et al., 2019). If *H. diminuta* can regenerate its GR, then A-P polarized patterns should be restored following amputation. We performed whole-mount *in situ* hybridization (WISH) for three posterior-enriched transcripts (*βcatenin1*, *βcat1*, WMSIL1_LOCUS14475; *frizzled5/8*, *fzd5/8*, WMSIL1_LOCUS9542; *tumor necrosis factor receptor-16*, *tnfr16*, WMSIL1_LOCUS10339) and one anterior-enriched transcript (*secreted frizzled related protein*, *sfrp*, WMSIL1_LOCUS14856) (Fig. 1E, uncut). The antisense riboprobes used showed distinct patterns while matched sense riboprobes showed minimal

background (Fig. S1). Worms were amputated within the GR (0.5 mm) and at 0 dpa all markers were reduced or lost (Fig. 1E). A-P polarized patterns were restored by at least 10 dpa, demonstrating normal GR patterning (Fig. 1E).

## GR regeneration fails after serial amputation

We investigated whether GR regeneration occurs after serial amputation. Newly harvested worms were acclimated *in vitro* for 3 days, and 0.5 mm anterior fragments were obtained (cut 1) and grown for 18 days before re-amputation (cut 2) (Fig. 2A). Subsets of each group were fixed to capture the variation and reproducibility of the amputations performed. We found that tissues extended slightly after heat-killing and fixation, resulting in slightly longer 0 dpa fragments: 0.63±0.07 mm and 0.61±0.16 mm after each amputation (Fig. 2B, means). Although fragment length did vary, there was no statistically significant difference at each time point. The GR lengths were longer at cut 1 (1.32±0.11 mm) than at cut 2 (1.09±0.08 mm), yet the percentage of GR length retained was close to the target range of 50% and the mean differences were not statistically significant (48±9% after cut 1 and 56±11% after cut 2) (Fig. 2C, means). When pooled together, there was a bias toward increased percentage of GR length retained after cut 2 (Fig. 2C, all), indicating that, despite our best efforts, the average amputation position in the GR may have been slightly different between the two time points.

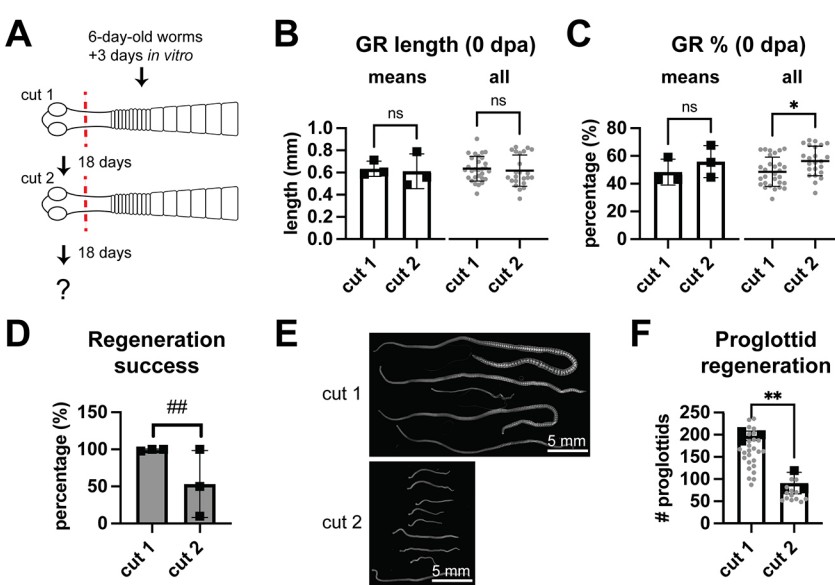

**Fig. 2. Serial amputation within the GR negatively affects regenerative ability.** (A) Scheme for serial amputation within the GR. (B,C) Quantification of GR lengths and percentage of GR length retained compared to GR lengths of uncut worms at 0 dpa per time point. Black squares represent group means, gray circles represent individual worms. $N=3$, $n=29, 23$; *$P<0.05$ (unpaired, two-tailed $t$-test). ns, not significant. (D) Quantification of regeneration success rate with $F$-test to compare variances ($^{\#\#}P=0.0015$). (E) DAPI-stained regenerates at 18 dpa. (F) Quantification of proglottids from worms that regenerated after cut 2. $N=3$; $n=64, 25$; **$P<0.01$ (unpaired, two-tailed $t$-test). Error bars represent s.d.

After cut 1, almost all fragments regenerated GRs and proglottids (Fig. 2D). After cut 2, many fragments failed to regenerate at all, producing inconsistent regeneration rates per experiment (Fig. 2D). Worms that did regenerate after cut 2 were significantly shorter (Fig. 2E) and produced fewer proglottids (Fig. 2F). Thus, GR regeneration can fail under certain conditions.

## Signals from the head regulate proglottid regeneration and GR maintenance

When amputating within the GR, the wound site is closer to the head. We suspected that the head asserts a negative effect on regeneration. After head amputation, regenerates exhibited increased growth and proglottid regeneration (Fig. 3A), indicating that signals from the head do inhibit growth. Using 1 h uptake of the thymidine analog (2′S)-2′-deoxy-2′-fluoro-5-ethynyluridine (F-*ara*-EdU) (Neef and Luedtke, 2011) to quantify the density of proliferating cells in the GR, we found that head amputation increased proliferation, which can explain the increased growth (Fig. 3B). In uncut worms, proliferation density in the GR increased with distance from the head (Fig. 3C), further supporting our conclusion that the head provides anti-proliferation signals. Upon head amputation and *in vitro* culture for 3 days, the increasing proliferation density along the GR A-P axis was no longer significant (Fig. 3D). However, the curves did not fully flatten. While signals from the head do regulate proliferation in the GR, the effects are slow to dissipate or mediated by additional factors resident within the GR.

Many regenerating animals respond to wounding by increasing proliferation locally. Following amputation in the GR, we quantified proliferation at both anterior- and posterior-facing wounds and found increased proliferation (Fig. S2). However, the local increase in proliferation was equivalent regardless of head presence (Fig. S2). Thus, amputation itself does not explain the increased proliferation in worms without a head but could contribute to increased proliferation.

Previous results have shown that GR maintenance depends on the head; after head amputation, the GR is eventually lost, the regenerates become fully proglottidized and cannot continue adding proglottids (Rozario et al., 2019). To further characterize how the head affects proglottid regeneration and GR maintenance, four amputation schemes were compared: +head, ½ head, −head and −0.5 mm anterior (Fig. 4A). Making transverse cuts through the head was technically challenging so a sampling of ½ head fragments were fixed

immediately and stained with anti-SYNAPSIN antibodies to visualize the nervous system. Although the amount of head tissue removed did vary, the brain was always retained (Fig. S3A, yellow arrowheads). After making the four different amputations at the anterior region, we assayed proglottid regeneration following serial amputation. All worms were amputated posteriorly to obtain fragments of the same length (2 mm), that were grown *in vitro* for 14 days, then re-amputated for three more cycles (Fig. 4A). The posteriors were collected to determine the number of proglottids regenerated after every cycle. As previously reported (Rozario et al., 2019), tapeworms regenerated after serial amputation when the head was present (Fig. 4B). The ½ head regenerates consistently displayed increased proglottid regeneration compared to +head regenerates (Fig. 4B). Both +head and ½ head regenerates maintained their GRs throughout 2 months of *in vitro* culture. When the head was amputated completely (−head), proglottid regeneration eventually ceased although some regenerates still had GRs after the fourth amputation (Fig. 4B). These −head regenerates had significantly shorter GRs (Fig. S3B), indicating that complete proglottidization was imminent. Previously, we observed rapid loss of the GR after head amputation, but anterior GR tissue was also removed in those experiments (Rozario et al., 2019). In this study, we were careful to make minimal head amputations that retained as much anterior GR and found that the GR persists for much longer than previously reported but does eventually recede. By contrast, fragments in which both the head and anterior GR were removed (−0.5 mm) became fully proglottidized by the second amputation or earlier (Fig. 4B). In summary, GRs were maintained if the brain region was retained but GRs were lost when additional anterior tissue was removed (Fig. 4C).

The number of proglottids regenerated also depended on amputations at the head, although our analysis was complicated by GR loss in −head and −0.5 mm fragments. After the first amputation, a clear dose-dependent response was detected: when more head tissue was retained, the number of proglottids decreased (Fig. S3C). This trend was maintained after serial rounds of amputation when comparing +head and ½ head regenerates, in which GRs were not lost (Fig. S3D). After the fourth round of amputation, the remaining head area plotted against number of proglottids for +head and ½ head regenerates revealed a negative correlation between head tissue abundance and proglottidization (Fig. 4D). We also measured the effect of head tissue abundance on proliferation within the GR using

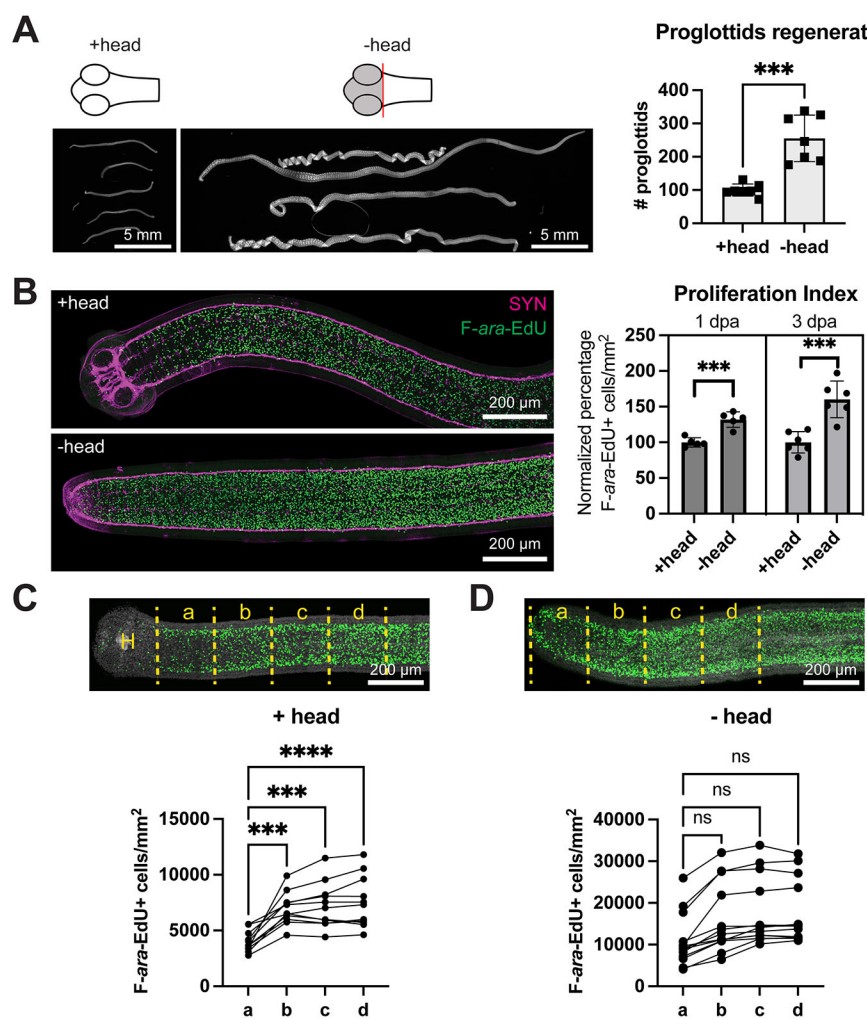

**Fig. 3. The head inhibits proglottid regeneration and cell proliferation.** (A) DAPI-stained regenerates from 0.5 mm amputations ±head at 15 dpa. Quantification of proglottids regenerated from a representative experiment. $n$=7; ***$P$<0.001 (unpaired, two-tailed $t$-test). (B) Confocal maximum intensity projections of a +head worm and a −head worm at 3 dpa immunostained for the neuronal marker SYN after a 1 h pulse of F-$ara$-EdU to label proliferating cells. Quantification of proliferation density within the GR from a representative experiment. Each sample was normalized to the mean proliferation density in the corresponding +head group. $n$=10, 12; ***$P$<0.001 (unpaired, two-tailed $t$-test). (C,D) Confocal maximum intensity projections after a 1 h pulse of F-$ara$-EdU ±head at 3 dpa. Yellow dotted lines represent 200-μm-wide regions used for quantification of proliferation density. H, head. $N$=3, $n$=11, 13; ***$P$<0.001, ****$P$<0.0001 (one-way ANOVA with Dunnett's multiple comparison test). ns, not significant. Error bars represent s.d.

F-$ara$-EdU. A general trend of increasing proliferation with decreasing head and anterior tissue emerged, although differences between +head and ½ head were not statistically significant (Fig. 4E,F). Taken together, the head effects on maintaining the GR and regulating proliferation can be separated. Signals from the brain region are necessary to maintain the GR, whereas proliferation and proglottidization are negatively regulated by head-dependent signals in a dose-dependent manner.

### $\beta$cat1 is necessary for proglottid regeneration and stem cell maintenance

To begin to ascertain the molecular regulators of GR maintenance, stem cell proliferation and proglottid regeneration rate, we turned to the Wnt signaling pathway. Wnt signaling determines A-P polarity in many systems (Petersen and Reddien, 2009) and plays roles in regulating cell proliferation, survival and differentiation (Nusse and Clevers, 2017). Expression of multiple members of the Wnt signaling pathway were found to be either anterior-enriched or posterior-enriched by RNA sequencing along the A-P axis of the GR (Rozario et al., 2019). As βcatenin protein is a key downstream effector of Wnt signaling, we decided to examine its function. *H. diminuta* has three *βcatenin* paralogs, as do other parasitic flatworms (Montagne et al., 2019). BLASTp comparisons of putative *H. diminuta* βcatenin paralogs with published sequences for other flatworms (*Schmidtea mediterranea*, *Schistosoma mansoni* and *Echinococcus multilocularis*) demonstrated that *Hdim-βcat1* (WMSIL1_LOCUS14475) is the

rat tapeworm ortholog of *βcat1* (Table S1). Expected conserved domains, including multiple armadillo repeats and the DSGxxSxxx[S/T]xxxS motif for CKI/GSK-phosphorylation necessary for targeting by the destruction complex (Montagne et al., 2019; Valenta et al., 2012), are present (Fig. S4A). Capacity to bind α-catenin cannot be ruled in or out at present (Fig. S4B; discussed below). Thus, *Hdim-βcat1* likely mediates Wnt signaling and may also function in cell adhesion.

In 6-day-old adults, *Hdim-βcat1* (hereafter *βcat1*) expression was largely absent in the head and anterior GR but increased toward the posterior GR (Fig. 5A). *βcat1* was broadly expressed in many tissues, in transverse stripes at proglottid boundaries and within accessory reproductive structures (Fig. S5). We performed RNA interference (RNAi) experiments by injecting double-stranded RNA (dsRNA) targeting *βcat1* throughout the GR and grew 2 mm anterior fragments *in vitro*. A dramatic growth defect was captured just 7 dpa/10 days post-injection (dpi) (Fig. 5B). Decreased growth was evident by 3 dpa and this RNAi treatment was ultimately lethal. The proglottids formed were extremely small and impossible to quantify but there was a marked decrease in worm lengths (Fig. 5C). RNAi of *βcat1* reduced transcript levels to 8.7 ±2.0% (Fig. 5D). In canonical Wnt signaling, *βcat1* is a transcription factor with many targets, including *axins*, though *bona fide* targets in tapeworms have not been determined. Tapeworm *axins* are highly divergent, but previous work using exogenous expression of AXIN1 and AXIN2 from the fox

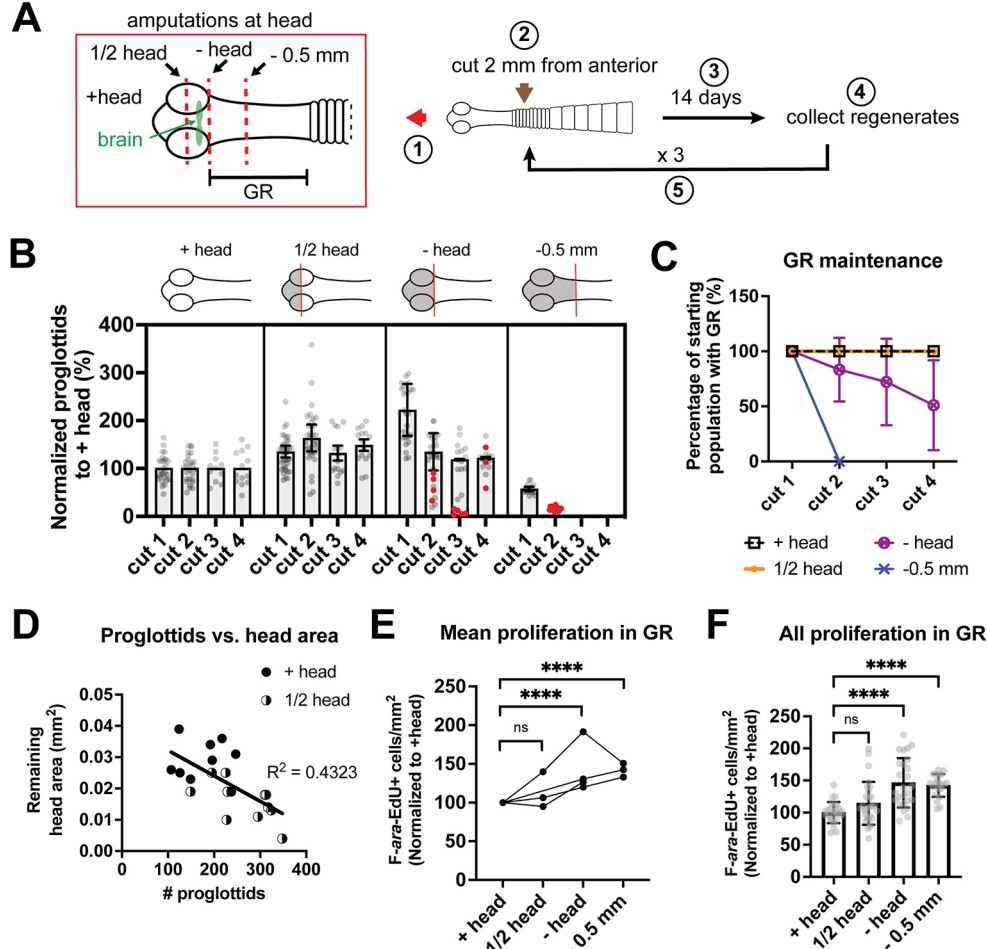

**Fig. 4. Effect of head or head-adjacent amputations on proliferation and regeneration.** (A) Scheme for results shown in B,C and Fig. S3. Step 1: Four amputation schemes were tested: intact head (+head), transverse cut through half the head while retaining the brain (½ head), head removal (−head) and cut 0.5 mm from the anterior into the GR (−0.5 mm). Step 2: All worms were amputated posteriorly to obtain 2 mm fragments. Step 3: Fragments were grown *in vitro* for 14 days. Steps 4, 5: Regenerates were re-cut to obtain 2 mm anterior fragments that were cultured *in vitro* while the posteriors were collected to measure proglottid regeneration. (B) Quantification of proglottids regenerated. All samples were normalized to the mean of +head worms set to 100%. Bars indicate means for all samples after cut 1 and only for samples with visible GRs for cuts 2-4. Proglottidized worms without GRs are indicated by red dots. *n*=27, 37, 26, 19 (+head, ½ head, −head, −0.5 mm). (C) Percentage of regenerates that maintained a visible GR from the data shown in B. (D) After cut 4, proglottids regenerated from +head and ½ head samples from one experiment plotted against head area remaining measured from DAPI-stained worms. Linear regression fitted; *n*=8, 12 (+head, ½ head). (E,F) Quantification of F-*ara*-EdU⁺ cells normalized to GR area at 3 dpa. All samples were normalized to the mean of +head worms set to 100%. (E) *N*=4, 3, 4, 3 (+head, ½ head, −head, −0.5 mm). ****$P<0.0001$ [two-way ANOVA (mixed model) and Tukey's multiple comparison test]. (F) *n*=26, 28, 26, 21 (+head, ½ head, −head, −0.5 mm). ****$P<0.0001$ (all pooled data from E were analyzed with one-way ANOVA compared to +head with Dunnett's multiple comparison test). ns, not significant. Error bars represent s.d.

tapeworm *E. multilocularis* showed evidence that both AXINs function canonically in the Wnt signaling pathway (Montagne et al., 2019). In *H. diminuta*, *axin1* (WMSIL1_LOCUS3986) was broadly expressed in overlapping patterns with *βcat1*, whereas *axin2* (WMSIL1_LOCUS6411) was highly tissue-specific (Fig. S5). After *βcat1* RNAi, *axin1* expression was markedly reduced to 18.8±11.1% of normal levels (Fig. 5D). This suggests that canonical Wnt signaling is inhibited following *βcat1* RNAi, although other roles for *βcat1* cannot be ruled out.

Given the strong decrease in growth after *βcat1* RNAi, we hypothesized that cycling stem cells require *βcat1*. Proliferation was significantly decreased in *βcat1* RNAi GRs compared to *GFP* RNAi controls (Fig. 5E). WISH for two known stem cell markers in *H. diminuta*, *minichromosome maintenance complex component-2* (*mcm2*; WMSIL1_LOCUS11918) and *laminB receptor* (*lbr*; WMSIL1_LOCUS1236) (Rozario et al., 2019), showed decreased expression after *βcat1* RNAi (Fig. 5F). Thus, *βcat1* is required to

maintain stem cells in the GR through regulation of proliferation and/or other mechanisms.

### GR maintenance requires *sfrp*

We hypothesized that *βcat1* activity may be counterbalanced by anterior-localized Wnt inhibitors. Previous RNA-sequencing analyses showed that *sfrp* expression is enriched in the anterior GR (Rozario et al., 2019). Like other tapeworms, *H. diminuta* has one true *sfrp* family member and a second highly divergent member, *sfrp-like* (WMSIL1_LOCUS8400), which was not investigated further in this study. Interpro (Blum et al., 2020) predicts that Hdim-SFRP retains both frizzled and netrin domains albeit with some variation (Fig. S6; discussed below). Since SFRP can inhibit Wnt signaling (Leyns et al., 1997; Rattner et al., 1997; Wang et al., 1997), we pursued its potential role in regeneration.

By WISH, *sfrp* expression was strikingly anterior-enriched with sparse expression in the head and strong expression throughout the

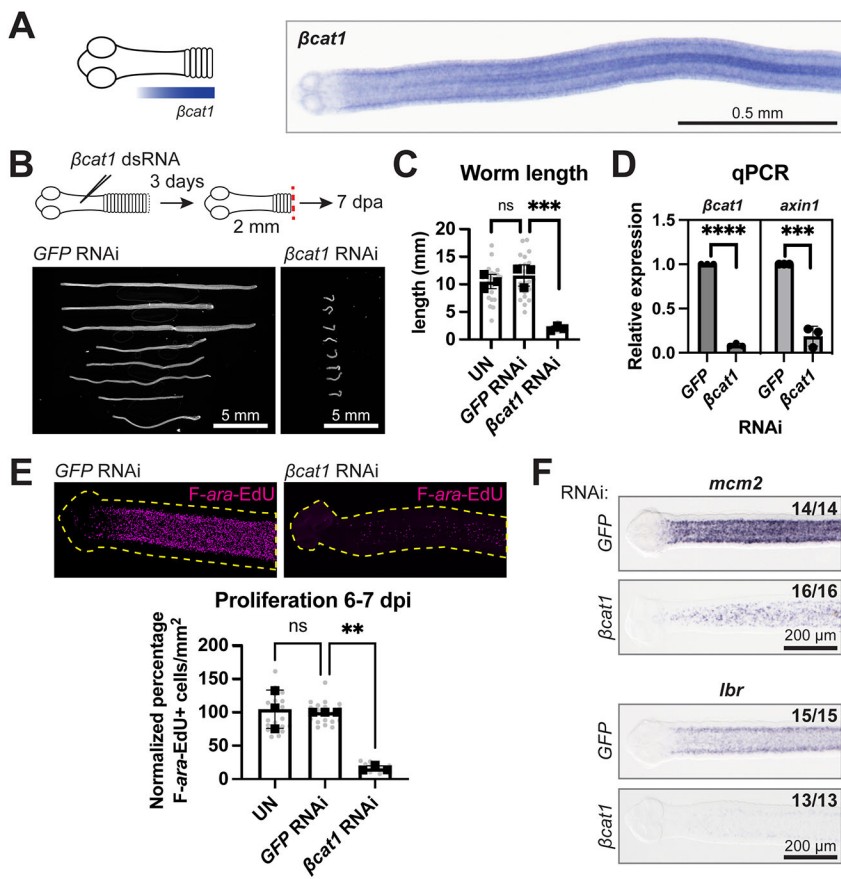

**Fig. 5. *βcat1* is a posterior-biased factor that is necessary to maintain the cycling stem cell population in the GR.** (A) Schematic and WISH of *βcat1* expression at the GR from a 6-day-old adult. (B) RNAi scheme and DAPI-stained worms at 7 dpa. (C) Quantification of worm lengths after RNAi at 7 dpa. Bars represent means, gray circles represent individual worms. *N*=3, *n*=24, 24, 23; \*\*\**P*<0.001 (one-way ANOVA with Dunnett's multiple comparison test). (D) qRT-PCR for target gene expression (*βcat1* or *axin1*) after *βcat1* RNAi compared to *GFP* RNAi (set at 1). *n*=3; \*\*\**P*<0.001, \*\*\*\**P*<0.0001 (unpaired, two-tailed *t*-tests). (E) Representative maximum intensity confocal projections at the GR after 1 h F-*ara*-EdU uptake. Dashed yellow lines delineate worm outlines. Quantification of F-*ara*-EdU⁺ cells normalized to area from a 800 µm-wide cropped region posterior to the head at 6-7 dpi. All samples were normalized to the mean of *GFP* RNAi worms set to 100%. Bars represent means, gray circles represent individual worms. *N*=3, *n*=20, 19, 16; \*\**P*<0.01 (one-way ANOVA with Dunnett's multiple comparison test). (F) WISH for the cycling stem cell markers *mcm2* and *lbr* at 6 dpi. *N*=3. ns, not significant; UN, uninjected. Error bars represent s.d.

GR that tapered off posteriorly (Fig. 6A). In adult tapeworms, *sfrp* expression was largely absent from the strobilated body except at lateral ends of proglottids (Fig. S5). We hypothesized that *sfrp* RNAi would promote proglottid regeneration as opposed to *βcat1* RNAi, which reduced proglottid regeneration. We injected *sfrp* dsRNA into the head, amputated 2 mm anterior fragments, re-injected dsRNA into the head and collected worms at 10-12 dpa (Fig. 6B). Contrary to our expectations, a modest decrease in worm lengths and proglottid regeneration were observed (Fig. 6C,D). Strikingly, the GRs were significantly shortened following *sfrp* RNAi (Fig. 6E,F). This is the first time GR length has been shown to be affected by RNAi of any target gene. Injecting dsRNA into the head does not preclude diffusion into the GR but the knockdown is likely to be anteriorly biased. We injected dsRNA throughout the GR and observed the same phenotypes (Fig. S7A-C). Both injection schemes resulted in similar *sfrp* knockdown efficacy (Fig. S7D).

Our results indicate that *sfrp* is necessary for proper GR maintenance but is unlikely to be involved in head-dependent inhibition of proliferation or proglottidization. Instead, *sfrp* could mediate head-dependent GR maintenance. Upon head amputation, we observed a modest 32.2±14.0% decrease in *sfrp* expression 3 days after head amputation (Fig. S7E). By WISH, the domain of *sfrp* expression after head amputation was often reduced as the GR shrinks, although it was highly variable at 3 dpa (Fig. S7F). We conclude that head amputation does not acutely abolish *sfrp* expression but may be required for long-term stable expression of *sfrp* at the GR.

Decreased proglottid regeneration after *sfrp* RNAi does not support the hypothesis that *sfrp* functions as a Wnt inhibitor. Proliferation within the GR was not significantly changed after *sfrp* RNAi (Fig. S7G). However, the shortening of the GR would bring the posterior boundary of the GR closer to the head. Thus, *sfrp* RNAi may indirectly exacerbate the anti-proglottidization effects from the head.

## Amputation distance from the head determines whether pro- or anti-regeneration signals dominate

Given these results, we hypothesized that the regeneration failure we observed after serial amputation within the GR (Fig. 2) was precipitated when the posterior wound was too close to the head, which allowed the growth inhibitory effects of the head to dominate. To test this, anterior fragments were allowed to regenerate after 0.5 mm amputation within the GR and then cut a second time at four different distances from the head. All samples were cut at ~1 mm, then subsets were progressively shaved closer toward the head (Fig. 7A, cuts A-D). Subsets of fragments were fixed at 0 dpa and we verified that cuts A-D progressively retained 87±5%, 64±7%, 47±9% and 22±4% of the GR, respectively (Fig. 7B). For each of three experiments, proglottid regeneration plummeted at either cut B or cut C (Fig. 7C). Groups with the lowest regeneration success corresponded to 0 dpa subsets where the posterior wound position clustered at ~60% GR length (Fig. 7C,D, matched colored arrows). This suggests that there is a tipping-point region within the GR that is refractory to regeneration following amputation. Surprisingly, the smallest fragments (cut D) regenerated equivalently to the longest fragments (cut A) (Fig. 7C,D, Fig. S7). Fragments from cut A and cut D achieved different absolute lengths at 18 dpa but the fold change in growth was comparable (Fig. 7E). There was no significant difference is GR length between regenerates from cut A or cut D (Fig. 7F). Thus, amputations within the GR can result in successful regeneration but not always, with more failed incidences when the wound site was at intermediate distances from the head. These results suggest that there

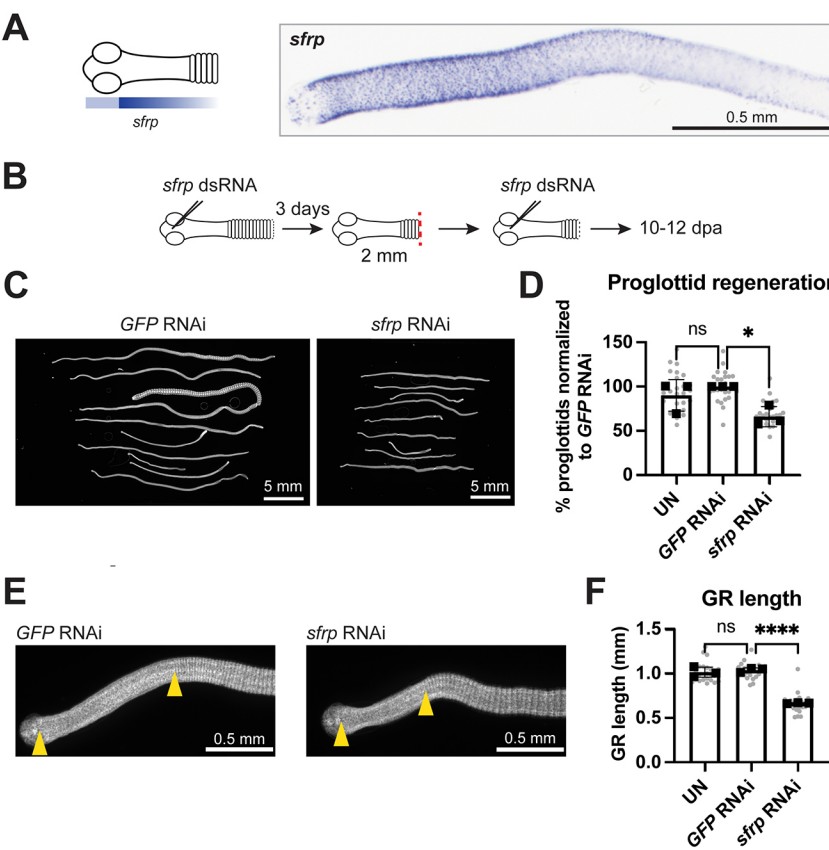

**Fig. 6. *sfrp* is an anterior-biased factor that regulates GR length.** (A) Schematic and WISH of *sfrp* expression at the GR from a 6-day-old adult. (B) RNAi scheme for the data shown in C-F. (C) Representative DAPI-stained regenerates after RNAi. (D) Quantification of proglottids regenerated after RNAi normalized to mean from *GFP* RNAi set to 100%. (E) Representative DAPI-stained images showing GR lengths (between yellow arrowheads). (F) Quantification of GR lengths from the data shown in D. (D,F) Bars represent means, gray circles represent individual worms. $N=3$, $n=26$, 25, 25; *$P<0.05$, ****$P<0.0001$ (one-way ANOVA with Dunnett's multiple comparison test). ns, not significant; UN, uninjected. Error bars represent s.d.

are yet more signals to be uncovered that explain how regeneration is promoted and restricted in tapeworms.

## DISCUSSION
### A working model for GR and proglottid regeneration

We propose the following working model (Fig. 8) to synthesize our current findings. The head provides signals that promote and restrict regenerative ability. These two roles can be separated anatomically and through molecular regulators. Proliferation and proglottidization rate are negatively regulated by the head in a dose-dependent and distance-dependent manner with increased proliferation farther from the head. Stem cells and germ cells are the only proliferative population in flatworms (Baguñà, 2012; Bolla and Roberts, 1971; Collins et al., 2013; Ishan et al., 2025; Koziol et al., 2014) and proglottid regeneration is a stem cell-dependent process (Rozario et al., 2019). We find that *βcat1* is expressed in a posterior-biased pattern at the GR and that RNAi of *βcat1* results in loss of proliferative stem cells. We speculate that the head may produce direct or indirect inhibitor(s) of unknown posterior signals that promote *βcat1* activity. Alternatively, the head-dependent inhibitors of regeneration may act independently of *βcat1*. By contrast, the head positively regulates regeneration by promoting GR maintenance. Without the brain region, the GR is gradually lost and when the anterior GR is also removed, GR loss happens quickly. RNAi of *sfrp* results in shortened GRs, indicating that *sfrp* contributes to GR maintenance. While *sfrp* expression is clearly anterior-enriched, it is unclear whether *sfrp* expression is promoted/maintained by signals from the head.

Taken together, our model provides an explanation for the highly mixed results in regeneration outcomes when re-amputations were made within the GR. When the posterior wound site was far enough from the inhibitors in the head, regeneration could occur. When the posterior wound site was close to the head, we speculate that the targets of the head-dependent inhibitors were not present. The head could promote proper growth of the GR, and at a sufficient distance from the head, proglottids regenerated. However, when the posterior wound site was at intermediate distances from the head, the inhibitors dominated leading to failure to regenerate. We speculate that the pro- and anti-regeneration contributions of the head result in a tipping point zone that is refractory to regeneration (Fig. 8). The exact players, direct and indirect relationships, as well as additional molecular mechanisms predicted by our working model still need to be tested. Alternatives to our model certainly exist. For example, GR and proglottid regeneration may occur from a wide range of posterior amputation sites with sporadic failures. These failures may be caused by anterior inhibitory signals that are not always sufficient to block regeneration. Future experiments will help determine whether regeneration in *H. diminuta* is largely stochastic or tightly regulated by spatially distinct molecular interactions.

Why GR regeneration did not fail significantly after the first round of amputations is unclear. Tapeworm dimensions change when grown *in vitro*. Despite acclimating worms *in vitro* for 3 days before all amputations, the spatial distribution and concentration of factors from the head to GR could have been different at cut 1 compared with cut 2 (Fig. 2). GR regeneration competence does not merely dissipate over time as the smallest fragments (cut D) successfully regenerated proglottids and the GR after the second round of amputation (Fig. 7). The head may be acting as an organizer that sets up a signaling environment within the GR that impacts regeneration positively and negatively in ways that we are just beginning to discover.

### Differing roles for *βcat1* in flatworm regeneration

In *H. diminuta*, *βcat1* is necessary to maintain stem cell abundance, and RNAi of *βcat1* results in a dramatic inhibition of growth, proglottid

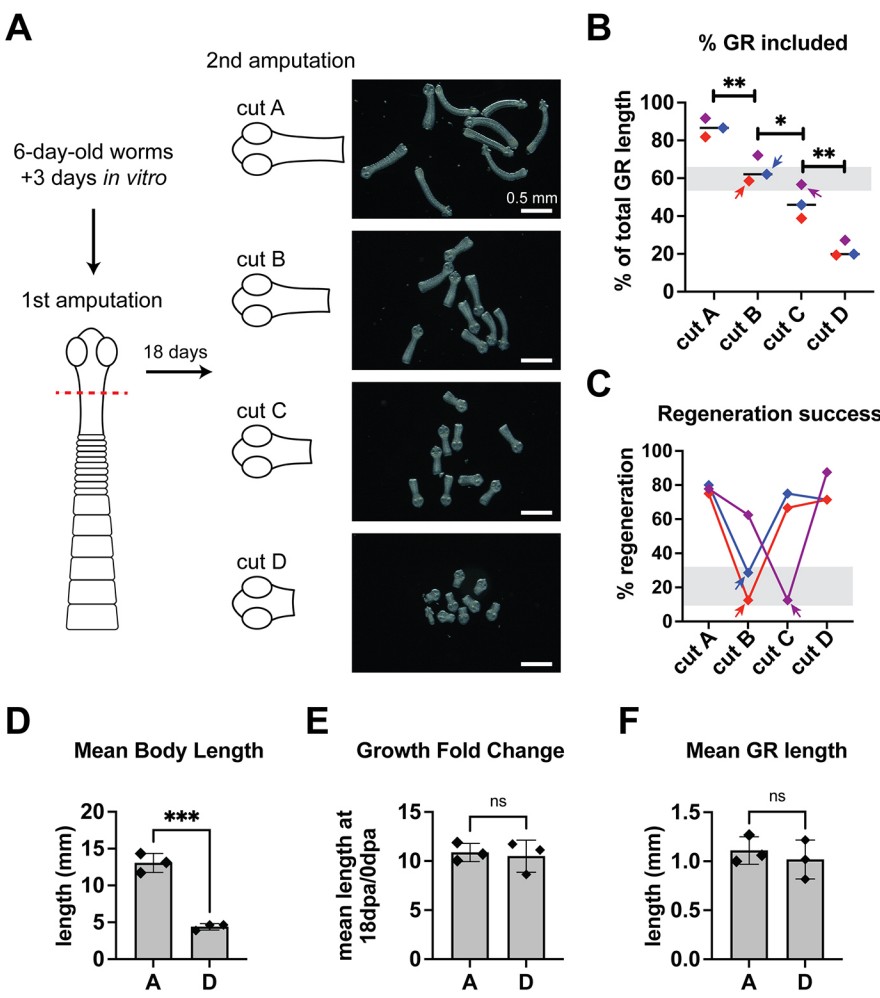

**Fig. 7. How amputations at increasing distance from the head influence competence to regenerate.** (A) Regenerates from 0.5 mm anterior fragments grown for 18 days were re-amputated at 1 mm (cut A). A subset was fixed to quantify GR length and another subset was grown *in vitro* for 18 days. The remaining fragments were re-cut to shave off ~20-25% of the GR (cut B) and the process repeated successively for cut C and cut D. Darkfield images of live fragments before *in vitro* culture are shown. (B) GR lengths measured from DAPI-stained 0 dpa fragments represented as percentage of GR length relative to uncut worms from the same cohort. $N=3$, $n=20, 21, 21, 21$; *$P<0.05$, **$P<0.01$ [two-way ANOVA (mixed model) with Tukey's multiple comparison test]. (C) Percentage of fragments that regenerated proglottids after 18 days. $N=3$, $n=27, 23, 21, 22$ at start. In B,C, each experiment is represented by the same color. Colored arrows point to matched sets representing groups with lowest regeneration success. (D-F) Quantification of worm length (D), growth fold change (E) and GR length (F). $N=3$, $n=20, 8, 10, 17$; ***$P<0.001$ (unpaired, two-tailed *t*-test). ns, not significant. Error bars represent s.d.

regeneration and ultimately death. In the liver fluke *Fasciola hepatica*, RNAi of *Fhep-βcat1* stunted juvenile growth, decreased proliferation and caused worm death (Armstrong et al., 2025), phenocopying our observations. Thus, roles for *βcat1* in regulating stem cell proliferation

**Fig. 8. Working model for regeneration in *H. diminuta*.** Overlapping regions at the head (blue bars) play two contrasting roles: proliferation and proglottidization rate are negatively regulated whereas GR maintenance is positively regulated. *βcat1* expression is posteriorly biased and *βcat1* is necessary to maintain stem cells in the GR, which are required for proglottid regeneration. Expression of *sfrp* is anteriorly biased and *sfrp* is necessary to maintain GR length. Signals that may link the head/brain/anterior GR to *βcat1* and *sfrp* expression or activity remain to be identified. Serial amputation within the GR revealed increased failure to regenerate the GR and proglottids (yellow cross) at intermediate distances forming a tipping point zone that is refractory to regeneration (black shading). Both smaller and larger fragments amputated outside of the black region regenerated equivalently (green ticks).

and/or survival may be a common theme in parasitic flatworms, as it is in many metazoans and in cancers (Mohammed et al., 2016; Steinhart and Angers, 2018).

In addition to canonical Wnt signaling, β-catenin proteins function in cell adhesion by linking cadherins with the actin cytoskeleton via α-catenin in many but not all species. In *S. mediterranea*, there is complete separation of function by two catenin paralogs: Smed-βCAT1 mediates Wnt signaling whereas Smed-βCAT2 mediates cell adhesion (Chai et al., 2010; Su et al., 2017). Accordingly, Smed-βCAT1 does not have α-catenin-binding sites (Chai et al., 2010; Su et al., 2017). The presence/absence of α-catenin-binding sites in βCAT1 from tapeworms is more ambiguous. Previously, a likely α-catenin-binding region demarcated by ten putative critical residues (Aberle et al., 1996; Pokutta and Weis, 2000) within a 26 amino acid span was identified for βCAT1 and βCAT2 in flatworms (Montagne et al., 2019). Compared to human βcatenin (CTNB1), we and others find strong conservation of this α-catenin-binding region in βCAT2 from multiple flatworms (Fig. S4B) (Montagne et al., 2019). While Smed-βCAT1 is clearly very divergent at this site, two tapeworms (*H. diminuta* and *E. multilocularis*) show more similarity to the human reference (Fig. S4B) (Montagne et al., 2019). Thus, we cannot rule out the possibility that βCAT1 in tapeworms functions in both Wnt signaling and cell adhesion, unlike their free-living counterparts.

The role of *βcat1* in posterior identity is well conserved across many taxa (Petersen and Reddien, 2009). In planarian species that are deficient in head regeneration, knockdown of *βcat1* rescues head regeneration, indicating that failure to regenerate the head in wild-

type worms was due to the inability to overcome the posteriorizing effects of *βcat1* (Liu et al., 2013; Sikes and Newmark, 2013; Umesono et al., 2013). *H. diminuta* cannot regenerate its head and we initially assumed that tapeworm *βcat1* would be acting in a homologous fashion to planarians. Because *βcat1* is required for stem cell maintenance in *H. diminuta*, potential roles in posteriorization could not be uncovered. RNAi manipulations of other Wnt signaling components that are beyond the scope of this study will help reveal a more holistic understanding of Wnt signaling in this system. However, tapeworms and planarians do differ in that *Smed-βcat1* does not appear to affect stem cells. Following RNAi of *Smed-βcat1*, there are no growth defects and normal blastemas form after amputation (Gurley et al., 2008; Iglesias et al., 2008; Petersen and Reddien, 2008).

The differences in utilization of *βcat1* between different species of platyhelminths is potentially very interesting. Perhaps segregating functions in posteriorization versus stem cell maintenance in *S. mediterranea* has enabled nimble remodeling of axes that makes this species so remarkable at regenerating from almost any fragment. Intestinal tapeworms need the suckers at the head to attach to the intestinal wall. Thus, maintenance of the head and proglottid regeneration may not be easily decoupled in tapeworms. Considering the wider role for βcatenin in regulating stem cell proliferation and survival across metazoa, it would be easy to assume that planarian phenotype is divergent. However, studies in the acoel *Hofstenia miamia*, a sister group to all bilaterians, show that RNAi of *Hmia-βcat1* is necessary for maintaining posterior identity but functions in regulating stem cells, proliferation or growth are not supported (Srivastava et al., 2014; Tewari et al., 2019). Thus, the planarian phenotype may be ancestral. A deeper understanding of how catenin paralogs have functionally specialized may reveal how strategies to regulate axial identity and stem cell proliferation/survival have evolved in regeneration-competent species.

### GR maintenance and the role of *sfrp*
The mechanism by which *sfrp* maintains the GR is unknown. While *sfrp* is primarily known as a Wnt antagonist, it can bind and stimulate frizzled receptors, interact with other signaling pathways and modulate cell-matrix interactions by binding integrins (Bovolenta et al., 2008; Mii and Taira, 2011). There is evidence that *sfrps* can stimulate Wnt and BMP signaling in the zebrafish retina (Holly et al., 2014). In *Drosophila*, *sfrps* can modulate Wnt signaling in a biphasic manner, inhibiting Wnt signaling when expressed at high levels while promoting Wnt signaling when expressed at low levels (Üren et al., 2000). In *H. diminuta*, the *sfrp* RNAi phenotypes were not the opposite of the *βcat1* RNAi phenotypes as one might predict if *sfrp* and *βcat1* were acting solely as Wnt antagonist and Wnt effector, respectively. In many flatworms, Wnt ligands are secreted from both anterior and posterior domains (Armstrong et al., 2025; Koziol et al., 2016; Reddien, 2018; Soria et al., 2020). Expression of *sfrp* throughout the GR may be required as a permissive signal to tune Wnt signaling to a moderately low level so that it can be acted upon differentially by other modulators. It is also possible that *sfrp* modulates *βcat1* activity differentially in a tissue- or cell-specific manner that we have yet to uncover. Alternatively, *sfrp* may function independently of Wnt signaling. Future studies investigating RNAi phenotypes of other antagonists, ligands and frizzed receptors will help resolve if or how other Wnt signaling components regulate GR maintenance.

Biochemical evidence for parasitic flatworm SFRP proteins as Wnt inhibitors has not been confirmed, although both frizzled and netrin domains are readily identifiable. There are conflicting assertions as to whether all six conserved cysteine residues typically found in the netrin domain are present (Armstrong et al., 2025; Koziol et al., 2016; Riddiford and Olson, 2011). We detect all six cysteine residues, but the sixth residue is displaced by three amino acids in flatworms, including free-living *S. mediterranea*, but not humans (Fig. S6). *Smed-sfrp1* is generally accepted as a Wnt inhibitor (Doddihal et al., 2024; Hill and Petersen, 2018; Reuter et al., 2015; Scimone et al., 2016; Stückemann et al., 2017; Sureda-Gómez et al., 2015). Thus, it remains possible that a muted or more ambiguous role in Wnt signaling by *sfrp1* is a shared feature in free-living and parasitic flatworms.

### Rationales for A-P polarized signals regulating regeneration
Our work has revealed that a balance of opposing signals at the GR can determine regeneration competence. It makes intuitive sense that initiation of proglottids must be inhibited close to the head if the GR is to be maintained as an unsegmented tissue. A recent study using the mouse bile duct tapeworm *Hymenolepis microstoma* has described discrete expression of Wnt and Hedgehog signaling components in a signaling quartet (SQ) within the transition zone from GR to fully elaborated proglottids (Jarero et al., 2024). It is possible that the SQ initiates proglottid formation and represents the unknown targets of head-dependent inhibitors predicted by our study. How the SQ responds to head-dependent signals and vice versa is currently unknown and warrants future investigation.

Our study also shows that regulation of proliferation dynamics is dependent on signals from the head. While amputation-induced proliferation does occur in *H. diminuta*, the enduring increase in proglottid regeneration after ½ head removal and dose-dependent effects of head tissue abundance on proliferation all suggest that the head provides signals that downregulate proliferation. What purpose do anti-proliferation signals serve in the GR? One intriguing possibility is that signals from the head could differentially regulate stem cell potency, self-renewal, asymmetric division or other cell cycle dynamics. There is already evidence for different cycling rates in stem cells of parasitic flatworms (Herz et al., 2024; Wang et al., 2018). Future investigation will help us understand how head-dependent signals may influence stem cell behaviors.

In short, the head and GR constitute a signaling environment that influences regeneration competence. Furthermore, the anterior GR contains a minimal cohort of stem cells and extrinsic signals that are required to regenerate the GR and seed new proglottids. Understanding these signals could enrich our knowledge of how stem cells and regeneration are regulated in parasitic flatworms and beyond.

### Study limitations
Our study relies on RNAi, which cannot eliminate gene expression. We also use *in vitro* culture conditions that necessarily deviate from conditions *in vivo*. How environmental signals influence our results is unknown. We have also only explored the contributions of two genes and tapeworm regeneration is likely to be a complex process involving interconnecting networks of interactions.

## MATERIALS AND METHODS
### Animal care and use
The life cycle of *H. diminuta* was maintained in house using mealworm beetles (*Tenebrio molitor*) and Sprague-Dawley rats for the larval and adult stages, respectively. Rats were fed 200-400 infective cysticercoids by oral gavage in ~0.5 ml 0.85% NaCl. Typically, adult tapeworms were collected 6 days after infection by euthanizing infected rats in a $CO_2$ chamber and

flushing the small intestine with 1× Hanks Balanced Salt Solution (HBSS; 21-023-CV, Corning). Rodent care was performed in accordance with protocols approved by the Institutional Animal Care and Use Committee (IACUC) of the University of Georgia (A2023 10-019-Y1-A0).

### In vitro parasite culture

Six-day-old tapeworms were harvested and acclimated to *in vitro* culture conditions for 3 days before any amputations were made. We used biphasic cultures as described previously (Rozario et al., 2019). 2.5× MEM essential amino acids (M5550, Sigma-Aldrich) was added to the liquid culture: Working Hanks 4 [WH4: 1× HBSS (MT21023CV; Corning), 4 g/l glucose (158968; Sigma-Aldrich), 1× antibiotic-antimycotic (15240062; Fisher Scientific)]. This ameliorated inconsistencies between batches of blood used for culturing but was not strictly necessary.

### Cloning genes for riboprobe and dsRNA synthesis

Cloning was performed as previously described (Collins et al., 2010). All primers and accessions are described in Table S2. Plasmids are available upon request. Briefly, adult *H. diminuta* cDNA was used to PCR amplify products that were ligated into pJC53.2 plasmid via TA cloning. Riboprobes for *in situ* hybridization were generated by *in vitro* transcription reactions with either T3 or SP6 RNA polymerase (Ishan et al., 2025). For RNAi, dsRNA was generated as previously described (Rouhana et al., 2013) using T7 RNA polymerase.

### In situ hybridization and other staining

Heat-kills were performed by swirling tapeworms in 75°C water for a few seconds before fixing [4% formaldehyde in PBS with 0.3% Triton X-100 (PBSTx)] for 30 min to 2 h at room temperature or overnight at 4°C. Worms were either dehydrated into methanol and stored at −30°C or used directly for staining. For simple phenotyping of growth and regeneration, worms were stained in 1 µg/ml DAPI (D9542, Sigma-Aldrich) in PBSTx overnight at 4°C and cleared in mounting solution (80% glycerol/10 mM Tris pH 7.5/1 mM EDTA) overnight before mounting and imaging. Previously published methods were employed for *in situ* hybridization (Rozario et al., 2019) and immunostaining (Rozario and Newmark, 2015). Anti-SYN antibodies (3C11; Developmental Studies Hybridoma Bank) were used at 1:200 with Tyramide Signal Amplification (TSA; Ishan et al., 2025; King and Newmark, 2013; Hopman et al., 1998).

### F-ara-EdU uptake and staining

F-ara-EdU pulses were performed in 0.1 µM F-ara-EdU (T511293, Sigma-Aldrich)/1% DMSO/WH4 at 37°C for 1 h. Staining was performed as previously described (Ishan et al., 2025) with the following specifics: (1) 15 min proteinase-K digestion (10 µg/ml in 0.1% SDS/PBSTx) and (2) 10 min TSA reaction. F-ara-EdU$^+$ cells were quantified using Imaris (Oxford Instruments) spot finder from 3D confocal *z*-stacks. Automatic spot discovery (with background subtraction) was enabled then manually adjusted to label all visible spots without false positives. All F-ara-EdU data are represented as densities by normalizing to the area of the quantified fields.

### Imaging and image processing

Confocal imaging was performed on a Zeiss LSM 900 with the following objectives: 20×/Plan-Apochromat/0.8 M27/FWD=0.55 mm and 63×/Plan-Apochromat/1.40 Oil DIC M27. WISH and DAPI-stained worms were imaged on a Zeiss AxioZoom V16 Microscope. Fiji (Schindelin et al., 2012) was used for brightness/contrast adjustments, maximum-intensity projections and measurements of lengths, areas, and numbers of proglottids.

### RNAi

Knockdowns were performed as previously described (Rozario et al., 2019). Tapeworms were microinjected with dsRNA (1-1.5 µg/µl in HBSS), and 2 mm anterior fragments were amputated 3 days later and cultured *in vitro*. Injections were performed using a Femtojet 4i (Eppendorf) at 500 hPa for 0.3-1 s. Injection of *GFP* dsRNA (Roberts-Galbraith et al., 2016) was used as a negative control. Experimenters were unaware of the dsRNA used throughout microinjections, staining and image analysis.

### qRT-PCR

To exclude genomic DNA, primers were designed to flank introns or span an intron-exon junction. Amplicon range was 133-277 bp. All primers were tested on cDNA ±reverse transcriptase and gel electrophoresis to confirm the presence of a single product. Primer efficiencies were calculated from twofold serial dilutions of cDNA. All primers are reported in Table S2.

For RNA extractions, 3 mm anterior fragments from four or five worms were cut, submerged in 200 µl TRIzol Reagent (15596026, Invitrogen) and processed according to manufacturer's instructions with these exceptions: (1) samples were homogenized with a motorized pestle twice in a semi-solid state then centrifuged at >12,000 *g* for 5 min to obtain 180 µl clean supernatant; (2) RNA pellets were washed twice with 75% ethanol-DEPC; and (3) pellets were resuspended in 40 µl nuclease-free water. DNase (M6101, Promega) treatment was performed for 1 h at 37°C. RNA was cleaned using RNA Clean & Concentrator-5 kit (R1016, Zymo Research) and the concentration measured on a NanoDrop spectrophotometer. cDNA synthesis was performed, according to the manufacturer's protocol, using Oligo(dT)$_{20}$ primers and the iScript Select cDNA Synthesis Kit (1708897, Bio-Rad) with undiluted RNA. cDNA was further diluted with 5-25 µl nuclease-free water depending on the number of reactions needed.

qRT-PCR was performed using GoTaq MasterMix (A6001, Promega) according to the manufacturer's instructions and scaled to 25 µl reactions with 2 µl cDNA template for each of three technical replicates. Each reaction had 0.4 µM primer pairs, 0.3 µM CXCR in 1× master mix. A 7500 Real Time PCR System (Applied Biosystems) was used for 40 amplification cycles: 15 s at 95°C and 1 min at 60°C. Relative gene expression change was calculated using the PFAFFL equation (Hellemans et al., 2007; Vandesompele et al., 2002) to account for different primer efficiencies. Internal normalization was carried out on the geometric mean of two previously published endogenous controls: *60S ribosomal protein L13* (*60Srpl13*; WMSIL1_LOCUS3984) and *myosin heavy chain* (*mhc*; WMSIL1_LOCUS8394) (Rozario et al., 2019).

### Statistical analysis

Graph Pad Prism 10 was used for all statistical analyses. All experiments were repeated at least twice. Error bars, statistical tests, number of replicates (*N*) and sample sizes (*n*) are indicated in corresponding figure legends and statistical significance indicated as *$P<0.05$, **$P<0.01$, ***$P<0.001$, ****$P<0.0001$. All raw data are available in Table S3.

### Acknowledgements

We are sincerely thankful to Phil Newmark and Melanie Issigonis (Morgridge Institute for Research, WI). Phil's unwavering support in the initialization of these projects and continued advice was instrumental in our success. We are also grateful for Melanie's critical evaluation of our manuscript and valuable feedback. We thank the Biomedical Microscopy Core at the University of Georgia, Athens, GA for confocal microscopy training and instrument maintenance as well as access to an Imaris workstation.

### Competing interests

The authors declare no competing or financial interests.

### Author contributions

Conceptualization: T.R.; Formal analysis: T.R., E.M.N.; Funding acquisition: T.R.; Investigation: T.R., E.M.N., L.E.P., I.R.S., T.H., M.F.C.; Supervision: T.R.; Writing – original draft: T.R., E.M.N.; Writing – review & editing: T.R., E.M.N., I.R.S.

### Funding

This work was funded by a National Institute of Allergy and Infectious Diseases (NIAID) grant (DP2 AI 154416-01 to T.R.). Open Access funding provided by the University of Georgia. Deposited in PMC for immediate release.

### Data and resource availability

All relevant data and details of resources can be found within the article and its supplementary information.

### Peer review history

The peer review history is available online at https://journals.biologists.com/dev/lookup/doi/10.1242/dev.204781.reviewer-comments.pdf

**Special Issue**
This article is part of the Special Issue 'Lifelong Development: the Maintenance, Regeneration and Plasticity of Tissues', edited by Meritxell Huch and Mansi Srivastava. See related articles at https://journals.biologists.com/dev/issue/152/20.

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
