## [Peer Review File · Development (Cambridge, England)]

Signals from the head and germinative region differentially regulate regeneration competence of the tapeworm *Hymenolepis diminuta*

Elise McCollough Nanista, Landon Elizabeth Poythress, Isabell Reese Skipper, Trevor Haskins, Marieher Felix Cora and Tania Rozario

DOI: 10.1242/dev.204781

Editor: Mansi Srivastava

Review timeline

Original submission:	10 March 2025
Editorial decision:	14 April 2025
First revision received:	10 July 2025
Editorial decision:	8 August 2025
Second revision received:	14 September 2025
Accepted:	15 September 2025

Original submission

First decision letter

MS ID#: dev.204781

MS TITLE: Anterior-posterior polarity signals differentially regulate regeneration-competence of the tapeworm *Hymenolepis diminuta*

AUTHORS: Tania Rozario; Elise McCollough Nanista; Landon Elizabeth Poythress; Isabell Reese Skipper; Trevor Haskins; Marieher Felix Cora

Dear Dr Rozario,

I have now received all the referees' reports on the above manuscript, and have reached a decision. The referees' comments are appended below, or you can access them online: please go to:

As you will see, the referees express considerable interest in your work, but have some criticisms and recommend a revision of your manuscript before we can consider publication. If you are able to revise the manuscript along the lines suggested, which may involve further experiments, I will be happy to receive a revised version of the manuscript. Your revised paper will be re-reviewed by one or more of the original referees, and acceptance of your manuscript will depend on your addressing satisfactorily the reviewers' major concerns. Please also note that Development will normally permit only one round of major revision. If it would be helpful, you are welcome to contact us to discuss your revision in greater detail. Please send us a point-by-point response indicating your plans for addressing the referees' comments, and we will look over this and provide further guidance.

Please attend to all of the reviewers' comments and ensure that you clearly highlight all changes made in the revised manuscript. Please avoid using 'Tracked changes' in Word files as these are lost in PDF conversion. I should be grateful if you would also provide a point-by-point response detailing how you have dealt with the points raised by the reviewers in the 'Response to Reviewers' box. If you do not agree with any of their criticisms or suggestions please explain clearly why this is so.

Reviewer 1

SUMMARY OF THE ADVANCE MADE IN THIS PAPER AND ITS POTENTIAL SIGNIFICANCE TO THE FIELD

In this manuscript, Nanista and colleagues explore the role of opposing signals that promote or inhibit regeneration of different tissues and regions in the tapeworm *Hymenolepis diminuta*. This organism has a head that is followed by the germinal region (GR), which is essential for posterior regeneration of the reproductive tissues (proglottids). The GR needs to be actively maintained, and under certain circumstances it can get depleted, leading to the failure to regenerate the more posterior tissues. The authors investigate the Anterior-Posterior signals and factors that maintain the GR and promote regeneration and inhibit proglottid formation. They first test the effect of the head region on GR maintenance and proglottid regeneration. Interestingly, they show that the head removal result in higher proglottid production, but at the expense of ultimate GR maintenance. This occurred if the brain region was removed during head removal, therefore the brain appears important in the mediation of GR maintenance and controlling proglottid formation. Authors then ask what molecular factors may be involved in these balancing processes. They hypothesize Wnt/Bcatenin to be a factor, and test Bcatenin RNAi. They find that beta-catenin is required for stem cell maintenance and upon Bcat RNAi, worms were significantly shorter than control worms, and stem cell markers showed a significant decrease. They also find that *sfrp1* is required for GR maintenance. Together their results suggest opposing signals present in the head and GR regions, providing a balance of GR maintenance versus proglottid production. Finally, they test different amputation distances from the head to see which one of these two opposing factors dominate in different regions. An unexpected finding is that the most severe amputation actually does not inhibit regeneration but produces similar results to the least severe amputation. Overall, this is a generally well-written manuscript, experiments and controls are carefully designed, carried out, and quantified, and exciting results are presented. However, at times the manuscript is hard to follow, and it will benefit greatly from editing for making it more accessible to a broader audience. This reader is left feeling excited about these results, but slightly unsatisfied and unclear about the main conclusions because some results were hard to synthesize and follow. I appreciate that the biology of the animal is quite complex, but I think there is room for authors to edit the text and figures to help making results clearer.

Major points

Introduction:

This is possibly the shortest introduction I have ever seen! Indeed, introduction suffers from being too short and therefore it was hard to follow. The authors need to work on the intro intensively to make some of the background information clear. It doesn't have to become very long, but it will help adding some details. See examples:

Line 57: I am already confused about the biology of the animal. When I compare this information with what's in Figure 1A, I end up getting conflicting messages: it looks like in Fig 1A, the animals cut in the proglottid region are regenerating proglottids???

I advise adding either a supplementary image, or an additional panel to Figure 1 showing the regeneration competent and incompetent regions/conditions clearly for the reader unfamiliar with this organism.

Line 62: I think this is confusing if the reader is not familiar with the system and I advice giving a couple more sentences explaining these important previous findings to setup the background more properly.

Results

GR regeneration fails after amputation - It would be nice if the authors can try to incorporate these findings into their model in Figure 8 and provide a more wholistic synthesis.

Line 127: very cool results

Line 137: But are there just fewer cells in that area to begin with? We need more detail in methods/figure legend. Did the authors make these counts relative to DAPI-positive cells?

Lines 227-288: I don't understand the rationale here. It feels reversed logic. They cite that Bcat is degraded by axin, but then they say that Bcat RNAi leads to decreased axin expression. This rationalization is confusing and need to be clarified.

Lines 241-242: Is this claim justified? So far, the authors show that the head (specifically the part including the brain) is required to maintain the GR. But I don't think data presented so far show that the head is playing a role in setting up gene expression in the GR to enrich Bcat posteriorly.

Seems like a stretch, and I suggest either clarifying or modifying this. I think the experiment to show this would have been making the head removal experiments like in Figure 4 but then looking at how this affects Bcat expression. I don't think this experiment has to be carried out, but without it, this claim is unfounded.

In fact they could simply start this section with stating Bcat plays a role but unclear through which mechanism, removing the speculative first sentence.

Line 256: "opposite phenotypes": I suggest spelling out these phenotypes to remind the reader.

Line 285: "...presence of head is necessary..." Finding this a bit of a stretch as well, since removing the head doesn't fully eliminate the sfrp expression, only reduces it.

Line 288: What's the rationale here for making double cuts instead of just one in these experiments?

Line 299: Did the GR regenerate in these animals? Can the authors be more explicit about these details?

Line 302 and after: This two paragraphs belong to discussion.

Line 306: Is this different than factor X?

Line 309: I suggest saying reduced instead of inhibited

Line 315: "Tapeworms grown in vitro..." What is the relevance of this sentence to the rest of this paragraph?

Discussion

I suggest adding subheadings to the discussion summarizing the points being made

Line 328: If the authors need space, this paragraph can be safely removed, and they can start this section with their model.

Methods

Line 494: Please provide information for where to find probe sequences (if in supplementary, please indicate). If the info not provided, please include details for what region of the coding sequence etc was used for probe design for each probe used in this study.

Line 516: We need more detail about how this quantification was made, especially whether it was a percentage of overall number of cells in that area, or whether they just counted EdU positive cells.

Line 519: If this is a previously-established method in this organism, please provide a citation.

Minor points

Line 353: italicize *Drosophila*

Figures

Figure 1

In the in situ panels, it would be helpful to mark GR and proglottids for the readers unfamiliar with this organism

Figure 2

based on the figure legend, I figured out that the ns for the graphs are 7-10, but from these figures they look like only 3 worms. I would suggest (but leaving it up to the authors) adding the ns underneath each bar for all figures, as this will strengthen the figures.

2E: why isn't there statistics here?

Figure 3

3A: suggesting making the head piece gray, highlighting the piece remaining, as in Figure 4B.

3C: Are there just fewer cells in this area? Do they account for that? (Also see my comments on this for the results section)

Figure 4

4A: should this schematic also indicate if/when GR is being removed (I think in -0.5mm?) - could be helpful to indicate

4B: should ½ head be -1/2 head? (with a minus sign)?

Figure 5

Please edit the figure schematics to show the experimental steps (like in Fig 6). When is RNAi injected, when is amputation carried out at what region?

5D: I would label this dotted line that it is GFP RNAi baseline, but I am a little confused about this panel. Is the dotted line for both Bcat and axin expression?

5E: Here it would be nice to know again whether they had DAPI staining, and whether there was overall a lower number of cells generally (not just EdUpositive cells)

Figure 8

Overall I appreciate this figure but it needs work to make it easier to follow. I am particularly having a hard time following B and C, and how they relate back to A.

8C: Can the authors be consistent in labeling the levels of the X factor across panels? B talks about it as HIGH, this one looks like it is exactly the same as A, but it should not be. Should it be labeled LOW? I also suggest avoiding red-green combination (not color blind friendly).

Reviewer 2

SUMMARY OF THE ADVANCE MADE IN THIS PAPER AND ITS POTENTIAL SIGNIFICANCE TO THE FIELD

The study by McCollough et al focuses on the functional delineation of a cohort of genes linked to Wnt signalling that control regeneration within the germinative region (GR) of the rat tapeworm, *Hymenolepis diminuta*. Firstly, the authors explore the ability of *H. diminuta* to regenerate the GR following amputation and demonstrate regenerative competence (although report highly variable results following serial amputations). Using RNAi, they find that β -catenin-1 is essential for the maintenance of stem cell proliferation, growth and survival, while SFRP regulates GR length. Relevant transcripts were also successfully localised using WISH.

The data presented build significantly upon those of Rozario et al (2019), are scientifically sound and provide molecular insight into a fundamental aspect of tapeworm biology. The value of the data lies within their potential application to other cestode species of medical importance with respect to therapeutic intervention. I deem the work of significant merit.

SUGGESTIONS TO AUTHORS

The manuscript is well written and the figures well presented. The manuscript does, however, require a little work in terms of structure - the results section is somewhat lengthy and could be more concise. To help with this Figure 8 (which is an excellent addition to the MS) could be moved to the discussion section which may be appropriate as it contains elements not yet confirmed experimentally or by data in MS.

L. 40 - Readers may benefit from a more detailed summary statement.

L. 84 - *In vitro* should be italicised - please correct here and throughout the manuscript.

L. 99 - *In situ* should be italicised - please correct here and throughout the manuscript.

L. 100-101 - The source in which these transcript loci were originally reported should be referenced after their mention.

L. 109 - "Is GR regeneration similarly robust?" - this style of question should be avoided. Perhaps change to "To investigate whether GR regeneration is similarly robust, we..."

L122-123 - 'The additional challenge of serial amputation revealed that GR regeneration does fail under certain conditions' either explain further or delete.

L. 126 - As above, direct questions like this should be avoided - best to rephrase here and elsewhere throughout the manuscript.

L. 170 - Reference? How long?

L136-137 - consider possibility that the effects are just slow to dissipate?

L. 157 - unclear as this could mean 2 mm from cut edge rather than original tip of head?

L 201 - unitalicise 'and'

L. 233-234 - Reduced proliferation does not necessarily correlate with an inability to maintain the somatic stem cell population - this would need investigated with an EdU/BrdU pulse-chase. Please rephrase.

L. 246 - "Like other tapeworms, *H. diminuta* has one true sfrp family member and a second highly divergent member" - please include a reference here as well as a gene ID for the second SFRP (WMSIL1_LOCUS8400).

L. 277 - should be Fig. S6D

L. 79 -85, 146-149, 283-287 - Introductions like this clutter an already complex, lengthy results section - may be best to remove.

L. 293-294 - '...GR and proglottid regeneration plummeted at cut B 294 or C over three independent experiments...' This is confusing as the graphical data show that 2 of the 3 replicates for cut C showed minimal (if any) reduction in regeneration?

L. 302-325 - As hypothetical, this appears out of place and adds no value to the results section. A much-reduced statement may be included here to enable the referencing of Figure 8, but please move the majority of this text to the discussion section.

- L. 329 - "The flatworm phylum" - Phylum Platyhelminthes.
- L. 335 - 341 - This section of text is unnecessary at the beginning of the discussion - please remove or incorporate into a conclusion.
- L. 375-384 - Large section of text describing Wnt antagonists in planarians - relate to results sooner.
- L. 199 - Incorrect spelling of mediterranea and Echinococcus.
- L. 347 - As the results of Bcat1 RNAi are presented before those of SFRP, it would be beneficial to the reader if the discussion was also structured in this order.
- L. 368 - in -> are
- L. 430-431 - reference?
- L. 440 - suppose -> propose?
- L. 509 - Please state concentration of proteinase K used.
- Figure 1D - While the staining does indeed appear to be specific, it would be beneficial to include images of negative control worms in the supplementary data.
- Figure 8 - Best suited to discussion.
- Table S1- Incorrect spelling of query (second column heading).
- Table S2 - Please include primers used to generate GFP dsRNA.

Reviewer 3

SUMMARY OF THE ADVANCE MADE IN THIS PAPER AND ITS POTENTIAL SIGNIFICANCE TO THE FIELD

This work investigates regeneration in the tapeworm *H. diminuta*. The germinative region (GR) is the only region where competence to regenerate exists. Prior work suggested that stem cells could be transplanted from different regions into the GR and function in regeneration, consistent with a model in which the signaling environment of the GR is required for regeneration rather than this region harboring a unique stem cell population with regeneration competence. This manuscript argues that signals from the head influence regeneration competence. In a wild result, removing the head resulted in more proliferation and much more growth than normal. beta-catenin was required, unexpectedly, for stem cell maintenance. sfrp was required for GR length and successful proglottid number regeneration. The results are intriguing, and suggest both positive and negative roles for the head in GR maintenance and proliferation regulation, but a clean model tying together all results is not yet emergent. Nonetheless, the work advances the field by finding important new pieces to the puzzle.

SUGGESTIONS TO AUTHORS

-In the text the experiment in Figure 1D could be more explicitly described - amputation at 0.5mm

-Not essential, but could add: intermediate time points between 0 and 10dpa to observe more detail on the return of patterned gene expression to the GR.

-Figure 2E could be explained more clearly. Is it that three separate experiments were performed with multiple animals each, and in one experiment 100% regenerated, in another experiment 50%, and the final experiment 10%? If that is correct, it is a strange form of variance, differing largely by trial more so than animal-to-animal. Any ideas about the difference between the different experiments? Regardless, it is clear that when regeneration did happen, fewer proglottids formed. In these cases, did the GR regenerate to its normal length (I imagine this could be determined from their existing pictures, as in Fig1D)? (I think the authors mean "widely" rather than "wildly" line 120)

-Figure 3: Figure 3B could be more clearly explained - was there not a second amputation (as in part A), but that the rest of the body was intact? On the other hand line 143 refers to the animals as "-head regenerates" leaving me to think they were amputated? In Figure S1 the authors found that proliferation 1dpa was similar at afw and pfw's, and they argue that the increase in proliferation at 3dp head amputation is therefore not likely explained simply as an increase in proliferation after head removal. This could also be improved if indeed the 3dpa data from 3B is from animals that were regenerating posteriorly. Regardless, it is difficult to eliminate the possibility from these data alone that the higher level of proliferation is simply caused by injury

rather than the loss of an inhibitory signal from the head. The authors could note this limitation in the text, but also point to the regeneration outcome in Fig3A that is consistent with their model of an inhibitory effect of the head. Another idea is to try removing a wedge of tissue from the GR to see if this fails to have the effect that removal of the head does, though trying this is not essential. The authors could return to address this caveat later in the text, more clearly pointing out the enduring affect of loss of half a head (Fig 4B) to the amount of proglottids regenerated in serial amputation, which is consistent with their model.

-Figure 4: Line 188 might want to add the word "negatively" to read "...are negatively regulated..."

-Figure 5: Axin can be a transcriptional target of Wnt signaling in multiple systems, and its expression by ISH could be assessed to complement the bcat ISH and RNAi work. The bcat1 RNAi result on stem cells is interesting and significant. Did the authors try to upregulate activity (with LiCl, GSK3beta inhibitors, or with RNAi of axin or APC?). Success with this is not essential for their paper to be able to make important claims about stem cell maintenance, but it could help support a model that bcat signaling levels influence proliferation rate. Similarly, or alternatively, wntless RNAi could support that it is Wnt signaling, rather than a role in adhesion, that explains the bcat1 role in stem cell maintenance.

-Figure 6: Does sfrp RNAi or serial amputation after sfrp RNAi lead to complete GR loss (not essential for conclusions)? I don't really understand the experiment in 6G - is it that just some of the region expressing sfrp is removed (so lower total levels per fragment), or is its expression level dynamically head dependent?

-Figure 7: line 286-88 - should it read?: "Perhaps if a posterior-facing wound were to be generated too close to the head, the growth inhibitory effect of the head would dominate and GR regeneration would fail." Why was this experiment done as a serial amputation experiment? Line 293-94: wording unclear, given amputation at site C (and site B) did not consistently result in regeneration failure, but sometimes did - wording could describe this result clearer. Given this variability it is not clear that cuts at D were different than B or C (3/3 seems not necessarily different from 2/3 or 1/3 resulting in regeneration?). Line 298 reads with interpretation: "...the inhibitory effect of the head was overcome when enough posterior GR tissue was removed". Wouldn't an alternative interpretation be that "...posterior regeneration, and regeneration of the GR, can happen at surprisingly anterior amputation planes, with sporadic failure. This indicates that any inhibitory signal from the anterior is not (always) sufficient to block posterior regeneration, even at amputation sites very close to the head."

-Figure 8: It is a bit speculative. I'm not convinced of the factor X in the manner that it is invoked (or that it would be graded in some way), or that if it exists it would act through bcat1. To my mind a simpler summary/model would be safer, which could be done graphically: 1. There is a "head" role in inhibiting proliferation and regeneration rate. 2. There is a head (or "head/anterior GR") role that promotes GR maintenance. 3. The GR can regenerate, with varied success, from amputation throughout the GR. 4. sfrp1 is required for GR length maintenance. 5. bcat1 is required for stem cell maintenance and regeneration. In such a cartoon, if the authors elected to such a variant, depicting the location of the brain, the GR location, the region inhibitory on proliferation, and the region promoting GR maintenance would be helpful (to the degree it can be depicted at present - graded color where boundary clarity is lacking could be added). It is up to the authors naturally to decide how they best want to present a summary/conclusions - but I think the paper would be stronger if some of the more speculative elements (with some restraint) were relegated to the text.

typo: medditerranea; Ecchinococcus

First revision

Author response to reviewers' comments

Rebuttal

We are immensely grateful for the positive reception to our manuscript and for the detailed and thoughtful suggestions made by all three reviewers. We have strived to make all the suggested changes where possible and have addressed each comment in blue below. We think that the revised manuscript is significantly improved by the changes suggested by the reviewers. Thank you.

In the revised manuscript, all changes to the text that directly address reviewer comments are highlighted in yellow. Yellow highlights also mark changes made to accommodate Development's standard for sample size reporting and any other errors. Changes made to adhere to the word limit are highlighted in blue. A study limitation section was not requested but we have added one. We are happy to remove/change it if the reviewers see fit.

Reviewer 1: SUMMARY OF THE ADVANCE MADE IN THIS PAPER AND ITS POTENTIAL SIGNIFICANCE TO THE FIELD

In this manuscript, Nanista and colleagues explore the role of opposing signals that promote or inhibit regeneration of different tissues and regions in the tapeworm *Hymenolepis diminuta*. This organism has a head that is followed by the germinal region (GR), which is essential for posterior regeneration of the reproductive tissues (proglottids). The GR needs to be actively maintained, and under certain circumstances it can get depleted, leading to the failure to regenerate the more posterior tissues. The authors investigate the Anterior-Posterior signals and factors that maintain the GR and promote regeneration and inhibit proglottid formation. They first test the effect of the head region on GR maintenance and proglottid regeneration. Interestingly, they show that the head removal result in higher proglottid production, but at the expense of ultimate GR maintenance. This occurred if the brain region was removed during head removal, therefore the brain appears important in the mediation of GR maintenance and controlling proglottid formation. Authors then ask what molecular factors may be involved in these balancing processes. They hypothesize Wnt/Bcatenin to be a factor, and test Bcatenin RNAi. They find that beta-catenin is required for stem cell maintenance and upon Bcat RNAi, worms were significantly shorter than control worms, and stem cell markers showed a significant decrease. They also find that *sfrp1* is required for GR maintenance. Together their results suggest opposing signals present in the head and GR regions, providing a balance of GR maintenance versus proglottid production. Finally, they test different amputation distances from the head to see which one of these two opposing factors dominate in different regions. An unexpected finding is that the most severe amputation actually does not inhibit regeneration but produces similar results to the least severe amputation. Overall, this is a generally well-written manuscript, experiments and controls are carefully designed, carried out, and quantified, and exciting results are presented. However, at times the manuscript is hard to follow, and it will benefit greatly from editing for making it more accessible to a broader audience. This reader is left feeling excited about these results, but slightly unsatisfied and unclear about the main conclusions because some results were hard to synthesize and follow. I appreciate that the biology of the animal is quite complex, but I think there is room for authors to edit the text and figures to help making results clearer.

Major points

Introduction:

This is possibly the shortest introduction I have ever seen! Indeed, introduction suffers from being too short and therefore it was hard to follow. The authors need to work on the intro intensively to make some of the background information clear. It doesn't have to become very long, but it will help adding some details. See examples:

Line 57: I am already confused about the biology of the animal. When I compare this information with what's in Figure 1A, I end up getting conflicting messages: it looks like in Fig 1A, the animals cut in the proglottid region are regenerating proglottids???

I advise adding either a supplementary image, or an additional panel to Figure 1 showing the regeneration competent and incompetent regions/conditions clearly for the reader unfamiliar

with this organism.

Line 62: I think this is confusing if the reader is not familiar with the system and I advice giving a couple more sentences explaining these important previous findings to setup the background more properly.

Thank you for this suggestion. We have added an introductory diagram with the basic tapeworm anatomy and a graphical summary of the most salient previous findings about how the tapeworm regenerates. We have removed the graph showing worms lengths every two days after amputation but retained the change in proglottid regeneration (Fig. 1D). The legend for Fig. 1 was adjusted to reflect these changes.

We have expanded the introduction to flesh out previous findings about *H. diminuta* regeneration that we hope will address the reviewer's concerns (Line 53-74).

The reviewer expressed surprise that the previous graphics looked like tapeworms cut in the proglottid region were regenerating proglottids. This is true. As long as the GR is present, new proglottids can regenerate. Fragments from the strobilated body that do not contain the GR cannot regenerate new proglottids.

Results

GR regeneration fails after amputation - It would be nice if the authors can try to incorporate these findings into their model in Figure 8 and provide a more wholistic synthesis.

Fig 8 is now changed to remove more speculative elements as suggested by Reviewer 3. The figure and text have been improved to address the incidences of GR regeneration failure (Line 293-333)

Line 127: very cool results
Thank you!

Line 137: But are there just fewer cells in that area to begin with? We need more detail in methods/figure legend. Did the authors make these counts relative to DAPI-positive cells? These data are taken from whole animal tissues. Quantifying DAPI+ cells is not feasible. The magenta dots (spots1) below were used to quantify F-ara-EdU+ (green) cells using Imaris. As you can see, it has high accuracy even for weak/deep signal within a 3D confocal z-stack. When we tried to quantify DAPI+(gray) cells using the same method, the blue dots (spots2) vastly underestimate the number of cells.

The proliferation data on all graphs in this manuscript are normalized to worm area to account for variations in tissue size. We have added language in the methods to better explain how we performed normalization (Line 488-493). We have published a detailed protocol for F-*ara*-EdU uptake, staining and quantification that is cited.

Visually, there is no apparent change in cell density at the anterior GR vs. any other region of the GR. The images below are from a single confocal slice showing a relatively even distribution of DAPI+ cells. After head amputation, there is an increase in F-*ara*-EdU+ cells and naturally, we would expect this to increase the total number of cells. However, we feel that normalizing to worm area is sufficient to account for variations in size from sample to sample but still represents the result that we see: increased density of proliferating cells in -head worms vs. + head worms.

Lines 227-288: I don't understand the rationale here. It feels reversed logic. They cite that Bcat is degraded by axin, but then they say that Bcat RNAi leads to decreased axin expression. This rationalization is confusing and need to be clarified.

The reviewer is right that this was confusingly written. We wanted to test if there was other evidence of compromised Wnt signaling following β cat1 RNAi. A direct way to do this would be to test for downregulation of β cat1 targets. No *bona fide* targets have been identified in tapeworms directly. *Axin2* is a common β cat1 target in many systems. Tapeworms have two *axin* paralogs and there is already evidence that both paralogs function canonically in Wnt signaling despite high divergence at the amino acid level, which we cite in the text. We find that in *H. diminuta*, *axin1* is broadly expressed in overlapping patterns with β cat1 whereas *axin2* is more sparse and tissue specific. Thus, *axin1* is a more likely putative target of β cat1. Indeed, we find that *axin1* expression is downregulated following β cat1 RNAi providing evidence that Wnt signaling is compromised. We do acknowledge that this observation does not rule out the possibility that β cat1 may also function independently of Wnt signaling. We have updated the text (Line 215-223) and added the WISH patterns for both *axins* in Fig. S5.

Lines 241-242: Is this claim justified? So far, the authors show that the head (specifically the part including the brain) is required to maintain the GR. But I don't think data presented so far show that the head is playing a role in setting up gene expression in the GR to enrich Bcat posteriorly. Seems like a stretch, and I suggest either clarifying or modifying this. I think the experiment to show this would have been making the head removal experiments like in Figure 4 but then looking at how this affects Bcat expression. I don't think this experiment has to be carried out, but without it, this claim is unfounded.

In fact they could simply start this section with stating Bcat plays a role but unclear through which mechanism, removing the speculative first sentence.

We agree with the reviewer and have removed the speculative statement. We have set up the introduction to this subsection with a statement hypothesizing that β cat1 activity may be counterbalanced by anterior-localized Wnt inhibitors (Line 234).

Line 256: "opposite phenotypes": I suggest spelling out these phenotypes to remind the reader.

We've changed this to describe expectations for proglottid regeneration as the phenotype (Line 246-247)

Line 285: "...presence of head is necessary..." Finding this a bit of a stretch as well, since removing the head doesn't fully eliminate the *sfrp* expression, only reduces it.

We agree with the reviewer and performed WISH for *sfrp* 3 days after amputation. The results show that the modest decrease in *sfrp* expression by qPCR may be an indirect consequence of GR shortening. What is clear is there is no obvious acute regulation of *sfrp* by the head. We have modified the description (Line 256-262), added new WISH data to the supplement (Fig. S7F).

Line 288: What's the rationale here for making double cuts instead of just one in these experiments?

Since we previously found that at cut 2, regeneration would fail at some frequency, we used the same scheme to test the hypothesis that regeneration failure was dependent on the distance of the posterior wound from the head. We have added clarifying language (Line 272-276).

Line 299: Did the GR regenerate in these animals? Can the authors be more explicit about these details?

Yes, the GRs regenerated. We've edited the description to emphasize the data that was provided in Fig. 7F (Line 286-287).

Line 302 and after: This two paragraphs belong to discussion.

We have moved all text regarding Fig. 8 to the discussion. (Line 291 and after)

Line 306: Is this different than factor X?

In the new Fig 8, we have removed factor X. We do discuss the possibility that there are posterior signals that regulate proglottidization, which may be inhibited by signals from the head (Line 302-305).

Line 309: I suggest saying reduced instead of inhibited

Thank you. In the revised text, this statement has been removed.

Line 315: "Tapeworms grown in vitro..." What is the relevance of this sentence to the rest of this paragraph?

The relevance is that all the tapeworm dimensions change *in vitro*. Though worms grown *in vitro* are able to regenerate and reproduce, they are not as big as they would be in their *in vivo* environment. Our standard practice is to acclimate worms to the *in vitro* environment for 3 days before making any amputations. We had previously noted this at the top of the results section (Line 89). We've decided to reiterate the point for clarity in the methods (Line 459-460). We've also added it directly to Fig. 2A and Fig. 7A. To reiterate: for all experiments using *in vitro* culture in this manuscript, "0 dpa" refers to 3 days post-acclimation from worms harvested 6 days post-infection.

The significance of this is that despite our efforts to acclimate the worms *in vitro*, the spatial distribution or concentration of signals within the GR at 0 dpa may not be exactly the same at cut1 vs. cut 2. We have compared the head area, GR area and mean/median width from fresh worms vs. worms cultured *in vitro* for 3 days and found no significant differences or even consistent trends. However, GR length does change and settles at ~1mm in length when worms are cultured *in vitro* (by the way, GR length also changes with time *in vivo*). This observation was previously published (Rozario et al., 2019). This is why we represented the amount of GR retained as a percentage. Though the difference in the means of %GR retained at cut 1 vs cut 2 was not significant, the pooled data does show a small increase in %GR retained at cut 2 (new Fig. 2C). It is difficult to know if this change in distribution is really a driving factor since there was a similar distribution in the variation of wound positions between cut 1 and 2.

We have added description about the differences between cut 1 and cut 2 to the text for Fig. 2

(Line 110-122). We have also clarified this point in the discussion (Line 327-333). Our data from Fig. 7 clearly shows evidence of a tipping point zone that is refractory to regeneration at cut 2. Why we didn't hit this zone at cut 1 is unknown. It is possible that at cut 1, the refractory zone is smaller and/or at a different location or that the effects of the head-dependent inhibitors could be more robustly overcome. We could spend enormous amounts of effort to try and chase down this difference by varying cut sites, cut times, acclimation time *in vitro* etc. but there will always be experimenter- and culture-induced variation. In many ways, serial amputation within the GR fortuitously revealed that inhibitory signals to regeneration exist at cut 2. We intend to continue exploring the signals that promote and inhibit regeneration by looking for molecular regulators.

Discussion

I suggest adding subheadings to the discussion summarizing the points being made
Thank you for this suggestion. We have done so.

Line 328: If the authors need space, this paragraph can be safely removed, and they can start this section with their model.

We have taken the reviewer's suggestion and removed this paragraph.

Methods

Line 494: Please provide information for where to find probe sequences (if in supplementary, please indicate). If the info not provided, please include details for what region of the coding sequence etc was used for probe design for each probe used in this study.

We had previously directed readers to Table S2 (primers, transcripts, and accessions) in the "Data and resource availability" section. Considering that this might be easy to miss, we removed the original language and improved the Methods section. A section describing cloning has been added and Table S2 is referenced here (Lines 468). We also direct readers to this table in the qPCR section (Line 514).

Redundant language was removed from the RNAi methods section.

Line 516: We need more detail about how this quantification was made, especially whether it was a percentage of overall number of cells in that area, or whether they just counted EdU positive cells.

These details are now provided (Line 486-493)

Line 519: If this is a previously-established method in this organism, please provide a citation.
The citation has been added (Line 503)

Minor points

Line 353: italicize
Drosophila
Corrected (Line 393)

Figures

Figure 1

In the in situ panels, it would be helpful to mark GR and proglottids for the readers unfamiliar with this organism

We have added arrows pointing to the head, GR and proglottids into one panel of Fig. 1E. The figure legend has been updated to orient readers.

Figure 2

based on the figure legend, I figured out that the ns for the graphs are 7-10, but from these figures they look like only 3 worms. I would suggest (but leaving it up to the authors) adding the ns underneath each bar for all figures, as this will strengthen the figures.

We neglected to specify that the points on the graphs are means from 3 independent experiments that each had 7-10 worms. The journal requires that we report sample sizes as exact numbers instead of ranges. For Fig. 2B-C, we now report the total number of worms across the 3 experiments for cut 1 and cut 2: n= 23, 27. We have added this to the legend.

We've also gone through all legends to standardize sample size and replication reporting.

2E: why isn't there statistics here?

This graph was intended to display that regeneration success was highly variable at the second cut. We've decided to add an F-test for differences in variance as it is a better statistical measure for this point over a comparison of means. The legend has also been updated.

Figure 3

3A: suggesting making the head piece gray, highlighting the piece remaining, as in Figure 4B. Thank you. We have made this change.

3C: Are there just fewer cells in this area? Do they account for that? (Also see my comments on this for the results section)

Addressed above.

Figure 4

4A: should this schematic also indicate if/when GR is being removed (I think in -0.5mm?) - could be helpful to indicate

4B: should $\frac{1}{2}$ head be $-\frac{1}{2}$ head? (with a minus sign)?

All 4 fragments retain some portion of the GR at the beginning. To clarify this point, we've added a line to denote the GR in Fig. 4A.

We've decided that $\frac{1}{2}$ head vs $-\frac{1}{2}$ head are not materially different.

Figure 5

Please edit the figure schematics to show the experimental steps (like in Fig 6). When is RNAi injected, when is amputation carried out at what region?

Figure 5B now includes the relevant schematic.

5D: I would label this dotted line that it is GFP RNAi baseline, but I am a little confused about this panel. Is the dotted line for both Bcat and axin expression?

The relative expression change compares expression for each target (*βcat1* or *axin1*) in *βcat1* RNAi worms compared to GFP RNAi worms (set at 1). To alleviate this confusion, we have modified the graph and legend to show the GFP RNAi comparison.

5E: Here it would be nice to know again whether they had DAPI staining, and whether there was overall a lower number of cells generally (not just EdUpositive cells)

Addressed above.

Figure 8

Overall I appreciate this figure but it needs work to make it easier to follow. I am particularly having a hard time following B and C, and how they relate back to A.

8C: Can the authors be consistent in labeling the levels of the X factor across panels? B talks about it as HIGH, this one looks like it is exactly the same as A, but it should not be. Should it be labeled LOW? I also suggest avoiding red-green combination (not color blind friendly).

This section has been moved to the discussion and modified. We have removed the more speculative elements as recommended by Reviewer 3. Factor X is not explicitly mentioned. We hope the new Fig. 8 and description (Line 293-335) are more straightforward.

Reviewer 2: SUMMARY OF THE ADVANCE MADE IN THIS PAPER AND ITS POTENTIAL SIGNIFICANCE TO THE FIELD

The study by McCollough et al focuses on the functional delineation of a cohort of genes linked to Wnt signalling that control regeneration within the germinative region (GR) of the rat tapeworm, *Hymenolepis diminuta*. Firstly, the authors explore the ability of *H. diminuta* to regenerate the GR following amputation and demonstrate regenerative competence (although report highly variable results following serial amputations). Using RNAi, they find that *β-catenin-1* is essential for the maintenance of stem cell proliferation, growth and survival, while SFRP regulates GR length. Relevant transcripts were also successfully localised using WISH. The data presented build significantly upon those of Rozario et al (2019), are scientifically sound

and provide molecular insight into a fundamental aspect of tapeworm biology. The value of the data lies within their potential application to other cestode species of medical importance with respect to therapeutic intervention. I deem the work of significant merit.

SUGGESTIONS TO AUTHORS

The manuscript is well written and the figures well presented. The manuscript does, however, require a little work in terms of structure - the results section is somewhat lengthy and could be more concise. To help with this Figure 8 (which is an excellent addition to the MS) could be moved to the discussion section which may be appropriate as it contains elements not yet confirmed experimentally or by data in MS.

L. 40 - Readers may benefit from a more detailed summary statement.

We have changed the summary statement:

This study explores contradictory roles of the tapeworm head in providing pro- and anti-regeneration signals, which can form a tipping point zone that is refractory to regeneration.

L. 84 - *In vitro* should be italicised - please correct here and throughout the manuscript. Changed all.

L. 99 - *In situ* should be italicised - please correct here and throughout the manuscript. Changed all.

L. 100-101 - The source in which these transcript loci were originally reported should be referenced after their mention.

We have made this change.

L. 109 - "Is GR regeneration similarly robust?" - this style of question should be avoided. Perhaps change to "To investigate whether GR regeneration is similarly robust, we...."

In the revised text, this question is no longer stated.

L122-123 - 'The additional challenge of serial amputation revealed that GR regeneration does fail under certain conditions' either explain further or delete.

We've changed this to "GR regeneration can fail under certain conditions" (Line 125)

L. 126 - As above, direct questions like this should be avoided - best to rephrase here and elsewhere throughout the manuscript.

To conform to the reviewer's preference, we removed all direct questions in the results section.

L. 170 - Reference? How long?

The reference for the original study is provided in the previous sentence. We can't measure the length difference in anterior GR retained between the two studies.

We suspect that L. 170 was a typo and the reviewer is referring to something in original Line 126-136. If the reviewer wanted clarification on how long the *F-ara*-EdU pulse period was, it was 1 hr. We have added this to the text to match the legend. A reference for *F-ara*-EdU use as a method to quantify cells in S-phase has also been added here instead of just in the methods. (Line 131)

L136-137 - consider possibility that the effects are just slow to dissipate?

We have added this possibility (Line 137-139).

L. 157 - unclear as this could mean 2 mm from cut edge rather than original tip of head?

We have changed this to: "All worms were amputated posteriorly to obtain fragments of the same length (2 mm), that were grown *in vitro* for 14 days, then re-amputated for 3 more cycles." (Line 156-157)

L 201 - unitalicise
'and'

Corrected

L. 233-234 - Reduced proliferation does not necessarily correlate with an inability to maintain the somatic stem cell population - this would need investigated with an EdU/BrdU pulse-chase. Please rephrase.

We used the term “maintenance” rather than self-renewal to distinguish between these two points. The term “maintenance” is agnostic to mechanism. If the stem cell population cannot be maintained, it could be due to defects in specification, self-renewal, survival or death. To alleviate any potential confusion, we have modified the concluding statement to: “Thus, *βcat1* is required to maintain stem cells in the GR through regulation of proliferation and/or other mechanisms.” (Line 230-231)

L. 246 - “Like other tapeworms, *H. diminuta* has one true *sfrp* family member and a second highly divergent member” - please include a reference here as well as a gene ID for the second SFRP (WMSIL1_LOCUS8400).

We have included this.

L. 277 - should be Fig. S6D

Thanks for that catch! It is now Fig. S7G.

L. 79 -85, 146-149, 283-287 - Introductions like this clutter an already complex, lengthy results section - may be best to remove.

We have removed those introductions except the first one, which has been simplified (Line 87-89)

L. 293-294 - ‘...GR and proglottid regeneration plummeted at cut B 294 or C over three independent experiments...’ This is confusing as the graphical data show that 2 of the 3 replicates for cut C showed minimal (if any) reduction in regeneration?

Both Reviewer 2 and 3 have pointed out how our initial description of this data is confusing. We have improved the text and Fig. 7B-C to highlight the main points. As we cannot track the same worms and match the % GR retained with the ability to regenerate, we have to make correlations with the final outcome and the % GR retained from subsets that were killed and fixed after amputation (0 dpa). For each experiment, there was a group where regeneration success plummeted; this group was either cut B or cut C. The 0 dpa subsets from those groups all cluster at the same point: ~60% GR retained. This supports our conclusion that there is a tipping point zone that is refractory to regeneration (Line 280-283).

L. 302-325 - As hypothetical, this appears out of place and adds no value to the results section. A much-reduced statement may be included here to enable the referencing of Figure 8, but please move the majority of this text to the discussion section.

As suggested, we have moved description of Fig. 8 to the discussion. We have also changed Fig. 8 to remove more speculative assertions as suggested by Reviewer 3. The new description is at Line 293-335.

L. 329 - “The flatworm phylum” - Phylum Platyhelminthes.

The relevant paragraph is now removed as was suggested by Reviewer 1.

L. 335 - 341 - This section of text is unnecessary at the beginning of the discussion - please remove or incorporate into a conclusion.

This paragraph was removed to accommodate the model discussion.

L. 375-384 - Large section of text describing Wnt antagonists in planarians - relate to results sooner.

We have shortened this paragraph to relate to the results sooner (Line 405-414).

L. 199 - Incorrect spelling of mediterranea and Echinococcus.

Corrected (Line 199).

L. 347 - As the results of *Bcat1* RNAi are presented before those of SFRP, it would be beneficial to the reader if the discussion was also structured in this order.

We have swapped the sections.

L. 368 - in -> are

This error is now removed from the new text (Line 406)

L. 430-431 - reference?

We amended this statement to better reflect the results from *βcat1* RNAi in planarians. Expression of stem cell markers and quantification of proliferation was not performed because the gross morphology does not suggest that stem cells are affected. Blastemas form after amputation and there is no growth defect. The references have also been added. (Line 369-372)

L. 440 - suppose -> propose?

We changed this to “assume” to be clearer (Line 380).

L. 509 - Please state concentration of proteinase K used.

We added the concentration (Line 489). The reference we provided is a methods paper that goes into great detail about the protocol. The timing of protK and TSA was specified because these conditions can be varied for specific applications.

Figure 1D - While the staining does indeed appear to be specific, it would be beneficial to include images of negative control worms in the supplementary data.

We have added the images of matching sense probes, which is now Fig. S1. Side-by-side development was performed and stopped at the same time. We biased toward over-development and found that all sense probes were clean.

Figure 8 - Best suited to discussion.

This has been moved to the discussion.

Table S1- Incorrect spelling of query (second column heading).

Corrected.

Table S2 - Please include primers used to generate GFP dsRNA.

Apologies for that oversight. The GFP primers and the reference are now included.

Reviewer 3: SUMMARY OF THE ADVANCE MADE IN THIS PAPER AND ITS POTENTIAL SIGNIFICANCE TO THE FIELD

This work investigates regeneration in the tapeworm *H. diminuta*. The germinative region (GR) is the only region where competence to regenerate exists. Prior work suggested that stem cells could be transplanted from different regions into the GR and function in regeneration, consistent with a model in which the signaling environment of the GR is required for regeneration rather than this region harboring a unique stem cell population with regeneration competence. This manuscript argues that signals from the head influence regeneration competence. In a wild result, removing the head resulted in more proliferation and much more growth than normal. beta-catenin was required, unexpectedly, for stem cell maintenance. sfrp was required for GR length and successful proglottid number regeneration. The results are intriguing, and suggest both positive and negative roles for the head in GR maintenance and proliferation regulation, but a clean model tying together all results is not yet emergent. Nonetheless, the work advances the field by finding important new pieces to the puzzle.

SUGGESTIONS TO AUTHORS

-In the text the experiment in Figure 1D could be more explicitly described - amputation at 0.5mm

We have clarified that the 0 dpa group in new Fig. 1E are 0.5 mm fragments (Line 106).

-Not essential, but could add: intermediate time points between 0 and 10dpa to observe more detail on the return of patterned gene expression to the GR.

Thank you for this suggestion. The process of establishing A-P polarity and patterning new segments is an unexplored area that we are actively pursuing. We are building a more specific suite of markers to capture how patterning is re-established during GR regeneration. We feel that this would be better explored in a different manuscript.

-Figure 2E could be explained more clearly. Is it that three separate experiments were performed with multiple animals each, and in one experiment 100% regenerated, in another experiment 50%, and the final experiment 10%? If that is correct, it is a strange form of variance, differing largely by trial more so than animal-to-animal. Any ideas about the difference between the different experiments? Regardless, it is clear that when regeneration did happen, fewer proglottids formed. In these cases, did the GR regenerate to its normal length (I imagine this could be determined from their existing pictures, as in Fig1D)? (I think the authors mean "widely" rather than "wildly" line 120)

Yes, the reviewer is right. Unfortunately, we can't track single worms and measure their GR at 0 dpa then follow them over time to see if they will regenerate. We have to cut worms then split the batch into two; kill and fix some for GR measurement and grow the rest *in vitro*. There's variation in the exact position of the amputation. Nonetheless, regeneration was robust after cut 1 but highly variable after cut 2. We've now represented more detailed graphs in Fig. 2B-C which show that while all batches were comparable, there is a slight increase in the % GR retained of cut 2 worms. At cut 2, we may have hit at a tipping point where regeneration will fail but because of variation in the position of the cut site, each batch had a highly variable number of successful regenerates. This tipping point was much better revealed in Fig. 7 when we purposefully staggered the cut site position. We've updated Fig. 7B-C to highlight the tipping point. We agree that this could have been better explained in the original submission. We've clarified the language now (Line 110-125).

Why we did NOT hit the tipping point when amputating within the GR initially is unclear. We address this in the discussion (Line 327-333). When tapeworms are grown *in vitro*, all their dimensions change. Though they are able to regenerate and reproduce, they are not as big as they would be in their *in vivo* environment. We acclimate the worms to the *in vitro* environment for 3 days before making any amputations to help ameliorate differences between early and late timepoints. The significance of this is that at 0 dpa for cut 1, the spatial distribution and/or concentration of signals within the GR may not be exactly the same as 0 dpa at cut 2. We have added more detailed explanations and updated Fig. 2 to compare cut 1 with cut 2 and though there is a slight increase in the % GR retained in the pooled fragments from cut 2, the two cuts were grossly equivalent. Still, it is possible that at cut 1, the refractory zone is smaller and/or at a different location because of the spatial distribution of signals within the GR. The concentration or distribution of head-dependent inhibitors could also be different. We could spend enormous amounts of effort to try and chase down this difference by varying cut sites, cut times, acclimation time *in vitro* etc. but there will always be experimenter- and culture-induced variation. Our study does suggest that there is unseen patterning in the underlying tissue that provides pro- and anti-regeneration signals. Going after the likely molecular regulators is how we would like to better address regeneration competence in *H. diminuta*. This paper represents the first step in addressing much larger questions that we will continue to investigate in future papers.

-Figure 3: Figure 3B could be more clearly explained - was there not a second amputation (as in part A), but that the rest of the body was intact? On the other hand line 143 refers the the animals as "-head regenerates" leaving me to think they were amputated?

There was no second amputation. Using "regenerates" was a mistake. We have changed it to "-head worms" (Line 145).

In Figure S1 the authors found that proliferation 1dpa was similar at afw and pfw's, and they argue that the increase in proliferation at 3dp head amputation is therefore not likely explained simply as an increase in proliferation after head removal. This could also be improved if indeed the 3dpa data from 3B is from animals that were regenerating posteriorly. Regardless, it is difficult to eliminate the possibility from these data alone that the higher level of proliferation is simply caused by injury rather than the loss of an inhibitory signal from the head. The authors could note this limitation in the text, but also point to the regeneration outcome in Fig3A that is consistent with their model of an inhibitory effect of the head. Another idea is to try

removing a wedge of tissue from the GR to see if this fails to have the effect that removal of the head does, though trying this is not essential. The authors could return to address this caveat later in the text, more clearly pointing out the enduring affect of loss of half a head (Fig 4B) to the amount of proglottids regenerated in serial amputation, which is consistent with their model.

We agree with the reviewer. Our data suggest that at least two things are occurring: local increase in proliferation at the wound site and suppression of proliferation by the head. While we like the idea behind the suggested experiment, removing tissue wedges is fraught with its own caveats and quite challenging to do in the GR where the worm width is ~200 μm . We agree that the enduring effect of increased proglottid regeneration after $\frac{1}{2}$ head removal and serial amputation argues against the idea that changes in proliferation we report can be solely driven by wound-induced proliferation.

We have adjusted the text to reflect that increased proliferation in -head worms could bear contribution from amputation-induced proliferation (Line 144-145). In the discussion, we address this caveat as suggested (Line 427-431).

-Figure 4: Line 188 might want to add the word "negatively" to read "...are negatively regulated..."

Added. (Line 186)

-Figure 5: Axin can be a transcriptional target of Wnt signaling in multiple systems, and its expression by ISH could be assessed to complement the *bcat* ISH and RNAi work. The *bcat* RNAi result on stem cells is interesting and significant. Did the authors try to upregulate activity (with LiCl, GSK3beta inhibitors, or with RNAi of axin or APC?). Success with this is not essential for their paper to be able to make important claims about stem cell maintenance, but it could help support a model that *bcat* signaling levels influence proliferation rate. Similarly, or alternatively, *wntless* RNAi could support that it is Wnt signaling, rather than a role in adhesion, that explains the *bcat* role in stem cell maintenance.

We are currently investigating the functions of multiple Wnt signaling components including negative regulators of the pathway. We appreciate all these suggestions and will address the role of canonical Wnt signaling vs. catenin-dependent adhesion in a separate study. We have also improved the text to clarify the decrease in *axin1* expression after *β cat1* RNAi, as suggested by Reviewer 1 (Line 215-221).

-Figure 6: Does *sfrp* RNAi or serial amputation after *sfrp* RNAi lead to complete GR loss (not essential for conclusions)? I don't really understand the experiment in 6G - is it that just some of the region expressing *sfrp* is removed (so lower total levels per fragment), or is its expression level dynamically head dependent?

In the time frame that these experiments were done, we did not see complete GR loss or complete suppression of *sfrp* expression for that matter. The reviewer is right that the change in *sfrp* expression following head amputation is too tenuous. The qPCR result reported was modest. We decided to perform WISH for *sfrp* 3 days after head amputation and found that the change in *sfrp* expression was variable and difficult to quantify objectively. The change we saw by qPCR may well have been driven by shortening GRs. Considering how our data in Fig. 4B shows that GR loss sometimes occurred over weeks after head amputation, we should have anticipated that any head-dependent regulation of *sfrp* would manifestly slowly and variably. We have changed the text to remove strong assertions about head-dependent *sfrp* expression (Line 257-262). The qPCR and WISH data showing *sfrp* expression after head amputation are now in the supplement (Fig. S7E-F). We have also softened our description in the model and discussion (Line 308-309).

-Figure 7: line 286-88 - should it read?: "Perhaps if a posterior-facing wound were to be generated too close to the head, the growth inhibitory effect of the head would dominate and GR regeneration would fail." Why was this experiment done as a serial amputation experiment? The language has been changed to: "Given these results, we hypothesized that the regeneration failure we observed after serial amputation within the GR (Fig. 2) was precipitated when the posterior wound was too close to the head, which allowed the growth inhibitory effects of the head to dominate." (Line 272-274).

Since we previously found that at cut 2, regeneration would fail at some frequency, we used the same scheme to test the hypothesis that regeneration failure was dependent on the distance of the posterior wound from the head.

Line 293-94: wording unclear, given amputation at site C (and site B) did not consistently result in regeneration failure, but sometimes did - wording could describe this result clearer. Given this variability it is not clear that cuts at D were different than B or C (3/3 seems not necessarily different from 2/3 or 1/3 resulting in regeneration?).

We have improved the explanation of this result in the figure and text (Line 280-283). In Fig. 7B-C, we have added arrows to highlight that the batches with the highest incidence of regeneration failure cluster around a similar wound position (~60% GR length). This suggests that there is a tipping point region that is refractory to regeneration.

Line 298 reads with interpretation: "...the inhibitory effect of the head was overcome when enough posterior GR tissue was removed". Wouldn't an alternative interpretation be that "...posterior regeneration, and regeneration of the GR, can happen at surprisingly anterior amputation planes, with sporadic failure. This indicates that any inhibitory signal from the anterior is not (always) sufficient to block posterior regeneration, even at amputation sites very close to the head."

We agree that this sentence is an interpretation rather than a description of the results. We have modified the text to clarify that amputations within the GR can result in successful regeneration but not always. We also observe that more failed regeneration incidences occur when the wound site was at an intermediate distance from the head (Line 286-289).

We have also added the reviewer's alternative explanation to the discussion (Line 321-323).

-Figure 8: It is a bit speculative. I'm not convinced of the factor X in the manner that it is invoked (or that it would be graded in some way), or that if it exists it would act through *bcat1*. To my mind a simpler summary/model would be safer, which could be done graphically: 1. There is a "head" role in inhibiting proliferation and regeneration rate. 2. There is a head (or "head/anterior GR") role that promotes GR maintenance. 3. The GR can regenerate, with varied success, from amputation throughout the GR. 4. *sfrp1* is required for GR length maintenance. 5. *bcat1* is required for stem cell maintenance and regeneration. In such a cartoon, if the authors elected to such a variant, depicting the location of the brain, the GR location, the region inhibitory on proliferation, and the region promoting GR maintenance would be helpful (to the degree it can be depicted at present - graded color where boundary clarity is lacking could be added). It is up to the authors naturally to decide how they best want to present a summary/conclusions - but I think the paper would be stronger if some of the more speculative elements (with some restraint) were relegated to the text.

Our goal with Fig. 8 was to present a working model to propose a potential explanation for the results, rather than to graphically summarize alone. However, the degree of speculation is high, and we agree with the reviewer that a more prudent approach is best. We also appreciate the detailed breakdown provided, which we have used as a guide. Fig. 8 has been replaced and the corresponding text has been moved to the discussion as requested by Reviewers 1 and 2. (Line 293-335)

typo: mediterranea; *Ecchinococcus*
Corrected (Line 199).

Second decision letter

MS ID#: dev.204781R1

MS TITLE: Anterior-posterior polarity signals differentially regulate regeneration-competence of the tapeworm *Hymenolepis diminuta*

AUTHORS: Tania Rozario; Elise McCollough Nanista; Landon Elizabeth Poythress; Isabell Reese Skipper; Trevor Haskins; Marieher Felix Cora

Dear Dr Rozario,

I have now received all the referees reports on the above manuscript, and am pleased to report that reviewers are by and large satisfied by your revision. The referees' comments are appended below, or you can access them online: please go to.

The overall evaluation is positive and we would like to publish a revised manuscript in Development, provided that the referees' comments can be satisfactorily addressed. Please attend to all of the reviewers' comments in your revised manuscript and detail them in your point-by-point response. If you do not agree with any of their criticisms or suggestions explain clearly why this is so. If it would be helpful, you are welcome to contact us to discuss your revision in greater detail. Please send us a point-by-point response indicating your plans for addressing the referees' comments, and we will look over this and provide further guidance.

Reviewer 1

SUMMARY OF THE ADVANCE MADE IN THIS PAPER AND ITS POTENTIAL SIGNIFICANCE TO THE FIELD

The authors have adequately addressed reviewer comments and improved the manuscript. I have no further suggestions and I thank the authors for their careful revision of this manuscript.

SUGGESTIONS TO AUTHORS

I have no further suggestions.

Reviewer 3

The paper still lacks a very conclusive mechanism, but contains a number of interesting observations that should advance the field. The presentation of data is improved. My remaining comments on this version are below:

Title: "Anterior-posterior polarity signals" - not sure this is the right term for the title

Intro: "We find two genes typically associated with Wnt signaling, sfrp and bcatenin1, as critical mediators of each head-dependent phenotype"

I find this sentence too strong, as it could be that bcatenin mediates proliferation independently of the role of the head in affecting proliferation; sfrp could similarly do something different than mediate the positive impact of the head, such as patterning/scaling of the GR.

Line 49: "...it is tempting to assume". I recommend changing - "one possibility is" or something else

Line 86 : I think this wording could be improved: "H. diminuta is competent to regenerate the GR."

"can" or "is capable of regenerating" or other

Fig 2D: states many fragments failed to regenerate at all and yet Fig 2F has no individuals with no proglottids. This could be clarified.

Its hard to see the 1/2 head line in Fig 4C

"Multiple members of the 194 Wnt signaling pathway were either anterior-enriched or posterior-enriched"

wording - it is the expression of these genes that is enriched.

Line 229: "showed decreased expression after beta cat1 RNAi". The authors could note if they saw fewer cells with similar signal intensity as WT, indicative of cell loss, or similar numbers of cells with weaker expression, indicative of decreased gene expression.

Lin 270: I had to read the paragraph and look at the figure a couple times to follow the wording. I still find this experiment unclear...not sure what there is to do about it, but maybe some wording tweaks could still be considered. Part of it is just that the result is hard to understand (variability and why cut D would do well).

Line 302: In principle the affects of the head need not act by inhibiting a posterior positive regulator of bcat - could in principle be completely independent; if the authors concur they could note this in the text.

Paragraph starting line 327 is somewhat hard to follow as worded

One idea for consideration: For sfrp - one possibility is it that it negatively regulates bcat in one tissue - increased posterior bcat activity could lead to loss of the posterior end of the GR (shrinking). Through some other Wnt expressed and affecting different cells (such as proliferative cells) bcat could affect stem cell maintenance and sfrp might not act on this wnt. One sfrp-affected bcat process and one non-affected bcat process in other words.

Paragraph starting on line 427 could be stated more succinctly, given little can be concluded from the present study about stem cells with different properties and the possible affect of the head on them - it seems a bit strange to end the paper on this line, as opposed to a more general conclusion from this work.

I think the working model in figure 8 is better than before

Second revision

Author response to reviewers' comments

Reviewer 1: SUMMARY OF THE ADVANCE MADE IN THIS PAPER AND ITS POTENTIAL SIGNIFICANCE TO THE FIELD

The authors have adequately addressed reviewer comments and improved the manuscript. I have no further suggestions and I thank the authors for their careful revision of this manuscript.

SUGGESTIONS TO AUTHORS

I have no further suggestions.

Thank you!

Reviewer 3: The paper still lacks a very conclusive mechanism, but contains a number of interesting observations that should advance the field. The presentation of data is improved. My remaining comments on this version are below:

Title: "Anterior-posterior polarity signals" - not sure this is the right term for the title

We have changed the title to:

Signals from the head and germinative region differentially regulate regeneration competence of the tapeworm *Hymenolepis diminuta*.

We have also adjusted the language in the abstract to address the reviewers concern with the original title:

Line 31-36:

In this study, we show that the head regulates regeneration competence by promoting maintenance of the GR and inhibiting proglottid formation in a distance-dependent manner. Anterior-posterior (A-P) patterning within the GR provide local signals that contribute to these responses. *bcat1* is necessary for stem cell maintenance, proliferation and proglottidization. On the other hand, *sfrp* is necessary for maintaining the GR at its proper length.

Intro: "We find two genes typically associated with Wnt signaling, *sfrp* and *bcatenin1*, as critical mediators of each head-dependent phenotype"

I find this sentence too strong, as it could be that *bcatenin* mediates proliferation independently of the role of the head in affecting proliferation; *sfrp* could similarly do something different than mediate the positive impact of the head, such as patterning/scaling of the GR.

We have softened the language as suggested:

Line 80-81:

We find two genes typically associated with Wnt signaling, *bcatenin1* and *sfrp*, as critical mediators of stem cell maintenance and GR length, respectively.

Line 49: "...it is tempting to assume". I recommend changing - "one possibility is" or something else

We have changed this to:

As a sister group to planarians, it is possible that shared regenerative abilities have enabled tapeworms to achieve their monstrous growth potential and reproductive prowess.

Line 86 : I think this wording could be improved: "H. diminuta is competent to regenerate the GR." "can" or "is capable of regenerating" or other

As suggested, we have changed it to:

H. diminuta can regenerate the GR.

Fig 2D: states many fragments failed to regenerate at all and yet Fig 2F has no individuals with no proglottids. This could be clarified.

To report the number of proglottids that regenerated after cut 2, we could only include worms that did regenerate. We feel the wording in the text is clear about that:

Line 124-125 (unchanged):

Worms that did regenerate after cut 2 were significantly shorter (Fig. 2E) and produced fewer proglottids (Fig. 2F).

For additional clarity, we modified the legend for 2F to reiterate the point:

Line 737-738:

(F) Quantification of proglottids from worms that regenerated after cut 2. N= 3; n= 64, 25; t-test. Error bars= SD.

Its hard to see the 1/2 head line in Fig 4C

We have modified the figure so that the +head line is spaced allowing the 1/2 head line to show through.

"Multiple members of the 194 Wnt signaling pathway were either anterior-enriched or posterior-enriched"

wording - it is the expression of these genes that is enriched.

We have changed the wording:

Line 193-195:

Expression of multiple members of the Wnt signaling pathway were either anterior-enriched or posterior-enriched by RNA sequencing along the A-P axis of the GR (Rozario et al., 2019).

Line 229: "showed decreased expression after beta cat1 RNAi". The authors could note if they saw fewer cells with similar signal intensity as WT, indicative of cell loss, or similar numbers of cells with weaker expression, indicative of decreased gene expression.

This is a qualitative result that readers can observe from the WISH images displayed. We make no claims about the number of marker+ cells or intensity per cell.

Lin 270: I had to read the paragraph and look at the figure a couple times to follow the wording. I still find this experiment unclear...not sure what there is to do about it, but maybe some wording tweaks could still be considered. Part of it is just that the result is hard to understand (variability and why cut D would do well).

We will trust that the readers will accept that surprising results and variability when analyzing organismal-level phenotypes are expected in science. We feel that the data showing the tipping-point region (which was distributed over subsets within cut B and C) is clearly presented by the matched colored arrows. To be as clear as possible, we have modified the wording in the text:

Line 279-282:

For each of three experiments, proglottid regeneration plummeted at either cut B or C (Fig. 7C). Groups with the lowest regeneration success corresponded to 0 dpa subsets where the posterior wound position clustered at ~60% GR length (Fig. 7C-D; matched colored arrows).

As for the surprising result that the smallest fragments (cut D) were competent to regenerate well, we agree that a complete explanation for this requires future studies. The concluding line of this paragraph captures this:

Line 289-290 (unchanged):

These results suggest that there are yet more signals to be uncovered that explain how regeneration is promoted and restricted in tapeworms.

Line 302: In principle the affects of the head need not act by inhibiting a posterior positive regulator of *bcat* - could in principle be completely independent; if the authors concur they could note this in the text.

We do agree with the reviewer. The original language invoked other Wnt inhibitors, crosstalk with other pathways and Wnt-independent mechanisms. To increase clarity, we have changed the language:

Line 302-304:

We speculate that the head may produce direct or indirect inhibitor(s) of unknown posterior signals that promote *βcat1* activity. Alternatively, the head-dependent inhibitors of regeneration may act independently of *βcat1*.

Paragraph starting line 327 is somewhat hard to follow as worded

To increase clarity, the language has been slightly modified:

Line 327-332:

Why GR regeneration did not fail significantly after the first round of amputations is unclear. Tapeworm dimensions change when grown *in vitro*. Despite acclimating worms *in vitro* for 3 days before all amputations, the spatial distribution and concentration of factors from the head to GR could have been different at cut 1 vs. cut 2 (Fig. 2). GR regeneration competence does not merely dissipate over time as the smallest fragments (cut D) successfully regenerated proglottids and the GR after the second round of amputation (Fig. 7).

One idea for consideration: For *sfrp* - one possibility is it that it negatively regulates *bcat* in one tissue - increased posterior *bcat* activity could lead to loss of the posterior end of the GR (shrinking). Through some other Wnt expressed and affecting different cells (such as proliferative cells) *bcat* could affect stem cell maintenance and *sfrp* might not act on this wnt. One *sfrp*-affected *bcat* process and one non-affected *bcat* process in other words.

We agree with the reviewer that this is a possibility. We have modified the discussion to include it:

Line 398-404:

Expression of *sfrp* throughout the GR may be required as a permissive signal to tune Wnt signaling to a moderately low level so that it can be acted upon differentially by other modulators. It is also possible that *sfrp* modulates *βcat1* activity differentially in a tissue/cell-specific manner that we have yet to uncover. Alternatively, *sfrp* may function independently of Wnt signaling. Future studies investigating RNAi phenotypes of other antagonists, ligands and frizzled receptors will help resolve if/how other Wnt signaling components regulate GR maintenance.

Paragraph starting on line 427 could be stated more succinctly, given little can be concluded from the present study about stem cells with different properties and the possible affect of the head on them - it seems a bit strange to end the paper on this line, as opposed to a more general conclusion from this work.

We have shortened this paragraph and added a more expansive conclusion section:

Line 432-442:

One intriguing possibility is that signals from the head could differentially regulate stem cell potency, self-renewal, asymmetric division or other cell cycle dynamics. There is already evidence for different cycling rates in stem cells of parasitic flatworms (Herz et al., 2024; Wang et al., 2018). Future investigation will help us understand how head-dependent signals may influence stem cell behaviors.

In short, the head and GR constitute a signaling environment that influences regeneration competence. Furthermore, the anterior GR contains a minimal cohort of stem cells and extrinsic signals that are required to regenerate the GR and seed new proglottids. Understanding these signals could enrich our knowledge of how stem cells and regeneration are regulated in parasitic flatworms and beyond.

I think the working model in figure 8 is better than before

Thank you.

Other comment:

We have removed the hyphen from “regeneration competence” as it is not necessary.

We thank the reviewers for all their helpful suggestions that have improved the manuscript.

Third decision letter

MS ID#: dev.204781R2

MS TITLE: Signals from the head and germinative region differentially regulate regeneration competence of the tapeworm *Hymenolepis diminuta*.

AUTHORS: Tania Rozario; Elise McCollough Nanista; Landon Elizabeth Poythress; Isabell Reese Skipper; Trevor Haskins; Marieher Felix Cora

ARTICLE TYPE: Research Article

Dear Dr Rozario,

I am happy to tell you that your manuscript has been accepted for publication in Development, pending our standard publication integrity checks.